# SELF-CONSISTENCY IMPROVES THE TRUSTWORTHINESS OF SELF-INTERPRETABLE GNNS

**Wenxin Tai**[†]**, Ting Zhong**[†]**, Goce Trajcevski**[§]**, Fan Zhou**[∗†‡]

[†] University of Electronic Science and Technology of China
[‡] Key Laboratory of Digital Media Technology in Sichuan Province
[§] Iowa State University

`wxtai@outlook.com, zhongting@uestc.edu.cn`
`gocet25@iastate.edu, fan.zhou@uestc.edu.cn`

## ABSTRACT

Graph Neural Networks (GNNs) achieve strong predictive performance but offer limited transparency in their decision-making. Self-Interpretable GNNs (SI-GNNs) address this by generating built-in explanations, yet their training objectives are misaligned with evaluation criteria such as faithfulness. This raises two key questions: (i) can faithfulness be explicitly optimized during training, and (ii) does such optimization truly improve explanation quality? We show that faithfulness is intrinsically tied to explanation self-consistency and can therefore be optimized directly. Empirical analysis further reveals that self-inconsistency predominantly occurs on unimportant features, linking it to redundancy-driven explanation inconsistency observed in recent work and suggesting untapped potential for improving explanation quality. Building on these insights, we introduce a simple, model-agnostic self-consistency (SC) fine-tuning strategy. Without changing model architectures, SC consistently improves explanation quality across multiple dimensions and benchmarks, offering an effective and scalable pathway to more trustworthy GNN explanations. Our code is publicly available at `https://github.com/ICDM-UESTC/SelfConsistencyXGNN`.

## 1 INTRODUCTION

Graph Neural Networks (GNNs) have achieved remarkable success across a wide range of tasks, from social network analysis (Wu et al., 2022a) to molecular property prediction (Wang et al., 2022). Despite their effectiveness, however, GNNs are often criticized as black boxes, which hinders their adoption in high-stakes and scientific domains (Pfeifer et al., 2022; Rajput & Singh, 2022; Warmsley et al., 2022). To address this issue, Self-Interpretable GNNs (SI-GNNs) (Velickovic et al., 2018; Lin et al., 2020; Sui et al., 2022; Miao et al., 2022) have been proposed, which jointly learn predictions and explanations in an end-to-end manner, making interpretability an intrinsic part of the model. This design not only makes decision-making more transparent but also yields structural insights that can facilitate causal discovery and scientific understanding.

Evaluating the quality of GNN explanations typically follows two paradigms (Agarwal et al., 2023). The first is explanation accuracy (Ying et al., 2019), which compares model-generated subgraphs against annotated ground-truth explanations. The second is faithfulness (Yuan et al., 2021), which evaluates whether the explanation correctly highlights the input elements that are truly relevant to the model's prediction (Azzolin et al., 2025a). Because faithfulness does not depend on ground-truth annotations, it can be applied more broadly across datasets and tasks, and has therefore become a widely adopted evaluation metric. However, there remains a mismatch between what we optimize and what we evaluate: SI-GNNs are trained with cross-entropy plus conciseness regularization (Tai et al., 2025), yet they are evaluated by faithfulness.

---

[∗]Corresponding author.

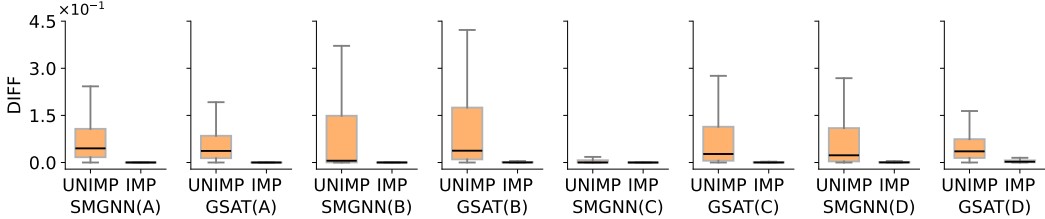

Figure 1: Self-inconsistency measured as the L1 difference (DIFF) between two explanations. Results are shown for SMGNN (Azzolin et al., 2025b) and GSAT (Miao et al., 2022) across four benchmark datasets: A = BA-2MOTIFS, B = MR, C = BENZENE, D = MUTAGENICITY. Within each panel, UNIMP and IMP denote unimportant and important features, respectively.

---

**This mismatch naturally raises two questions:**
*(i) Can faithfulness, as a conceptual property, be optimized during training?*
*(ii) Even if feasible, is it necessary—does it truly improve explanation quality?*

---

To answer the first question, a common evaluation of faithfulness (Amara et al., 2022) is to feed the identified explanation graph subset back into the model: if the prediction remains unchanged, the explanation is deemed faithful. This procedure implicitly relies on the SI-GNN being *self-consistent*. If the explanation is truly decisive, the SI-GNN should consistently highlight the same graph subset upon re-use. Consistent explanations in turn lead to consistent predictions, thereby satisfying the faithfulness criterion. This perspective suggests that faithfulness can indeed be optimized by introducing an alignment loss that enforces agreement between successive explanations.

Answering the second question is more subtle. While faithfulness can be optimized, it is not obvious whether doing so truly improves explanation quality. To investigate, we conduct an empirical study and find that, without faithfulness-oriented training, the first-pass and second-pass explanations of an SI-GNN can differ significantly. Using benchmark datasets with ground-truth explanations, we further observe that self-inconsistency primarily arises from instability on features labeled as unimportant, while important features remain stable (Figure 1). This observation resonates with recent findings (Tai et al., 2025) on explanation inconsistency across repeated training runs of the same SI-GNN, where inconsistency was traced to redundancy—explainers irresponsibly assigning high importance to unimportant features when budget allows. That work further showed that addressing redundancy improves explanation quality. Since our study shows that self-inconsistency also concentrates on unimportant features, it suggests a potential connection: addressing self-inconsistency may address redundancy as well, thereby improving explanation quality in a similar manner.

In this work, we propose to incorporate self-consistency into the training objective of SI-GNNs. On top of the standard objectives, we add a self-consistency (SC) loss that minimizes discrepancies between successive GNN explanations (Section 3.1). This simple yet general addition requires no modifications to the model architecture, and can be seamlessly integrated into existing SI-GNNs. We show that SC training drives feature importance scores toward a small set of near-fixed levels (Section 3.2), while conciseness regularization governs which levels are selected (Section 3.3). This interaction reshapes the explainer's behavior, resulting in more responsible and stable importance assignments. Extensive experiments across representative SI-GNNs and benchmark datasets demonstrate that SC consistently improves the trustworthiness of GNN explanations in terms of consistency, accuracy, faithfulness, and informativeness.

## 2 PRELIMINARIES

### 2.1 SELF-INTERPRETABLE GNNS (SI-GNNS)

We first formalize the task and model setup. A graph $G = (\mathcal{V}, \mathcal{E})$ consists of a set of nodes $\mathcal{V}$ and a set of edges $\mathcal{E}$. In graph classification, a GNN can be decomposed into a GNN encoder $h_Z : G \to \mathbb{R}^d$ that maps $G$ to a representation, and a classifier $h_{\hat{Y}} : \mathbb{R}^d \to \mathbb{R}^c$ that produces the prediction, giving $f = h_{\hat{Y}} \circ h_Z$. A SI-GNN adds an explainer $h_{G_s}$, yielding $f = h_{\hat{Y}} \circ h_Z \circ h_{G_s}$, where $h_{G_s}$ selects

a graph subset $G_s \subseteq G$ as the explanation. Following prior work (Tai et al., 2025), we focus on structural features (edges) in the instance-level setting.

**Definition 1** (Learning Objective). Given $(G, Y)$, the objective is to select a subset $G_s$ of $G$ that maximizes the mutual information (MI) with the label while promoting explanation conciseness:

$$\max_{G_s \subseteq G} I(G_s; Y) - \beta \cdot R(G_s), \tag{1}$$

where $I(\cdot; \cdot)$ denotes MI, $R(G_s)$ is an conciseness regularizer, and $\beta$ controls its strength. The form of $R(G_s)$ differs across SI-GNNs, and we review the most common ones below.

Existing SI-GNNs can be broadly categorized into four groups—attention-based, causal-based, size-constrained, and MI-constrained—depending on their subset selection strategy (Tai et al., 2025).

**(1) Attention-based methods** leverage attention mechanisms (Vaswani et al., 2017) to assign importance scores directly, without explicit regularization to enforce explanation conciseness. A representative example is GAT (Velickovic et al., 2018), which directly assigns importance scores to edges via attention mechanisms. The model is trained solely with classification loss:

$$\mathcal{L}_{\text{GE}} = \mathcal{L}_{\text{CE}}(Y, \hat{Y}|G_s) \tag{2}$$

where $\mathcal{L}_{\text{CE}}$ is cross-entropy loss.

**(2) Causal-based methods** employ causal inference techniques (Pearl, 2014) to identify causal patterns beyond spuriosity-prone statistical correlations. A representative example is CAL (Sui et al., 2022), which employs disentanglement and intervention to identify causal edges:

$$\mathcal{L}_{\text{GE}} = \mathcal{L}_{\text{CE}}(Y, \hat{Y}|G_s) + \beta \cdot \mathbb{D}_{\text{KL}}(\mathbb{P}_{\boldsymbol{\theta}}(\bar{Y}|\bar{G}_s)||\mathbb{Q}(\bar{Y})) + \gamma \cdot \mathcal{L}_{\text{CE}}(Y, \hat{Y}'|G_s \cup \bar{G}'_s), \tag{3}$$

where $\beta$ and $\gamma$ are pre-defined hyperparameters, $\bar{G}_s = G \setminus G_s$ is the complement of $G_s$, and $\bar{G}'_s$ represents the result of intervening on $\bar{G}_s$ (e.g., replacing its latent representations with those from other batch samples). $\mathbb{Q}(\bar{Y})$ is often set to a uniform distribution.

**(3) Size-constrained methods** enforce explanation conciseness by penalizing the size or sparsity of the selected subset. A representative example is SMGNN (Azzolin et al., 2025b), which adds a sparsity loss term to encourage concise and human-understandable explanations:

$$\mathcal{L}_{\text{GE}} = \mathcal{L}_{\text{CE}}(Y, \hat{Y}|G_s) + \beta \cdot \frac{|G_s|}{|G|}, \tag{4}$$

where $|G_s|$ and $|G|$ denote the number of edges in the selected subset and the original graph.

**(4) MI-constrained methods** minimize the MI between the selected subset and the original graph, providing an information-theoretic route to enforce explanation conciseness. A representative example is GSAT (Miao et al., 2022), which minimizes the MI between $G_s$ and $G$ via KL divergence:

$$\mathcal{L}_{\text{GE}} = \mathcal{L}_{\text{CE}}(Y, \hat{Y}|G_s) + \beta \cdot \mathbb{D}_{\text{KL}}(\mathbb{P}_{\boldsymbol{\theta}}(G_s|G)||\mathbb{Q}(G_s)), \tag{5}$$

where $\mathbb{Q}(G_s)$ is a Bernoulli distribution. Other SI-GNNs are discussed in Section A.

## 2.2 Explanation Redundancy

Recent work (Tai et al., 2025) has revealed that explanations produced by SI-GNNs are often inconsistent: when the same model is retrained with different random seeds, the explanations for a given instance may vary substantially. Through analysis, this inconsistency was attributed to *explanation redundancy*, as current SI-GNNs impose insufficient conciseness constraints, leaving excess budget that allows high importance scores to be irresponsibly assigned to unimportant edges. For instance, in GAT no explicit regularization is enforced, making redundancy unavoidable in principle; CAL avoids false negatives (missing important edges) but does not penalize false positives (including unimportant edges), thereby failing to discourage redundancy; SMGNN controls subset size via a sparsity coefficient $\beta$, but this is typically set small to balance accuracy and conciseness, weakening the constraint and allowing redundancy to persist; and GSAT faces the same issue—although it theoretically minimizes MI, in practice its authors report that retaining 50–80% of edges yields the best accuracy, which is far too many to eliminate redundancy.

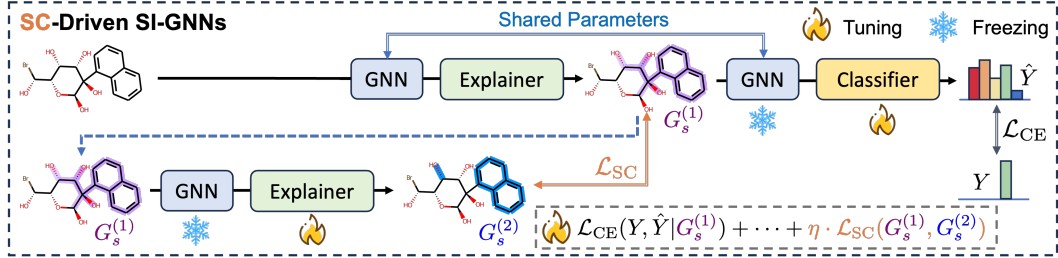

Figure 2: Overview of the SC training framework. A SI-GNN is first trained with the standard loss (Section 2.1). The encoder is then frozen, and the explainer and classifier are fine-tuned with an additional alignment loss that enforces the first-pass explanation $G_s^{(1)}$ (from $G$) to match the second-pass explanation $G_s^{(2)}$ (from $G_s^{(1)}$).

This phenomenon can be formalized under a unified budget-constrained formulation:

$$\max_{G_s \subseteq G,\ |G_s| \leq M} I(G_s; Y), \tag{6}$$

where $M$ denotes the maximum allowed subset size (implicitly determined by $R(G_s)$). Define $G_s^*$ as the ground-truth explanation subset. When $M \geq |G_s^*|$, there exist $\sum_{n=0}^{M-|G_s^*|} \binom{|G-G_s^*|}{n}$ distinct subsets that achieve the same maximum MI. This multiplicity of valid subsets makes it possible for unimportant edges to be included without affecting prediction accuracy, thereby formalizing why insufficient conciseness regularization naturally gives rise to redundancy in existing SI-GNNs.

While Tai et al. (2025) has focused on cross-model explanation inconsistency, our study reveals a related form of instability—explanation self-inconsistency within a single SI-GNN—which also occurs on unimportant edges. This motivates us to explicitly enforce self-consistency during training to reduce explanation redundancy and thereby improve explanation quality.

## 3 METHOD

### 3.1 SELF-CONSISTENCY TRAINING FRAMEWORK

Our framework consists of two steps. We begin by revisiting the workflow of a standard SI-GNN, which forms the basis for the first step. A GNN encoder first updates node representations:

$$\mathbf{V} = \text{GNN}(G). \tag{7}$$

An explainer, typically a multilayer perceptron (MLP), then predicts edge importance scores:

$$w_{ij} = \text{MLP}([\mathbf{v}_i; \mathbf{v}_j]), \quad \alpha_{ij} = \sigma(w_{ij}), \tag{8}$$

where $[\cdot; \cdot]$ denotes vector concatenation and $\sigma$ denotes the sigmoid function. Each edge $e_{ij}$ is included in the explanation subset $G_s$ with probability $\alpha_{ij}$, and we adopt the Gumbel–Sigmoid trick (Jang et al., 2017) to enable differentiable sampling. The selected subset $G_s$ is then processed by the same GNN encoder and pooled into a graph-level embedding, which is finally passed to an MLP (classifier) for prediction. Vanilla SI-GNNs are optimized using a combination of cross-entropy and conciseness regularization (Section 2.1); see Section B for implementation details.

**Step 1: Pretraining.** We first train the SI-GNN with the standard objective until convergence, and then freeze the GNN encoder. This ensures that representation learning is not disrupted by the additional SC loss introduced later, so that SC loss only influences the explainer's behavior.

**Step 2: Self-Consistency Fine-Tuning.** Given a graph $G$, the explainer produces an explanation $G_s^{(1)}$. We then feed $G_s^{(1)}$ back into the model, obtaining a second explanation $G_s^{(2)}$. If $G_s^{(1)}$ truly captures the decisive structure, the explainer should reproduce it consistently (i.e., $G_s^{(1)} = G_s^{(2)}$). To enforce this, we introduce a SC loss to minimize the discrepancy between successive explanations:

$$\mathcal{L}_{\text{SC}} = |G_s^{(1)} - G_s^{(2)}|. \tag{9}$$

The final training objective is the combination of the standard SI-GNN loss and $\mathcal{L}_{\text{SC}}$:

$$\mathcal{L}_{\text{FT}} = \mathcal{L}_{\text{GE}} + \eta \cdot \mathcal{L}_{\text{SC}}, \tag{10}$$

where $\eta$ is a pre-defined hyperparameter. The proposed two-step SC training framework (illustrated in Figure 2) is simple yet general: it requires no modification to the model architecture, and can be seamlessly applied to a wide range of existing representative SI-GNNs.

Define the second-pass mapping on each edge score as:

$$T(\alpha) \;=\; \sigma\big(g(\alpha)\big), \quad \alpha \in [0, 1], \tag{11}$$

where $g(\cdot)$ denotes the explainer's pre-activation on the second pass (obtained through the GNN encoder and explainer), and $\sigma$ is the sigmoid function. Intuitively, enforcing explanation self-consistency is equivalent to requiring $T(\alpha) \approx \alpha$, i.e., edge scores remain stable across passes. We next formalize this idea through the notion of near fixed levels.

## 3.2 Near Fixed Levels Induced by Self-Consistency

We formalize the concentration to a few levels without assuming the specific form of $g(\cdot)$.

**Definition 2** (Near fixed level). Given tolerance $\varepsilon \in (0, 1)$, $\alpha \in [0, 1]$ is an $\varepsilon$-near fixed level of $T$ if $|T(\alpha) - \alpha| \leq \varepsilon$. We denote the set by $\mathcal{S}_\varepsilon = \{\alpha : |T(\alpha) - \alpha| \leq \varepsilon\}$.

**Window condition.** For any candidate level $a^* \in (0, 1)$, the near fixed condition is equivalent to:

$$|T(a^*) - a^*| \leq \varepsilon \iff g(a^*) \in \Big[\text{logit}(a^* - \varepsilon), \, \text{logit}(a^* + \varepsilon)\Big], \tag{12}$$

where $\text{logit}(p) = \ln \frac{p}{1-p}$ denotes the logit function. Thus, achieving self-consistency requires the explainer's pre-activation $g(a^*)$ to fall inside a logit window around $a^*$.

**Window size.** The width of this admissible window is:

$$\Delta_g(a^*, \varepsilon) = \text{logit}(a^* + \varepsilon) - \text{logit}(a^* - \varepsilon) \approx \frac{2\varepsilon}{a^*(1 - a^*)}, \quad (\varepsilon \text{ small}). \tag{13}$$

For interior points $a^* \in (0, 1)$, the window is finite, and hitting such levels depends on model architecture and optimization trajectory. When $a^* \to 0$ or $a^* \to 1$, the two-sided window degenerates into a one-sided threshold:

$$\text{near } 1: \; g(a^*) \geq \tau(\varepsilon), \qquad \text{near } 0: \; g(a^*) \leq -\tau(\varepsilon), \tag{14}$$

where $\tau(\varepsilon) = \ln \frac{1-\varepsilon}{\varepsilon}$. These half-space conditions correspond to the flat saturation regions of the sigmoid, making $0/1$ relatively easier to achieve than interior levels.

Below, we analyze from a gradient perspective how, with adequately strong conciseness regularization (CR), SC can improve explanation quality.

## 3.3 Interaction with Conciseness Regularization

Consider the joint training loss:

$$\mathcal{L}_{\text{FT}} = \mathcal{L}_{\text{CE}} + \beta \cdot R(G_s) + \eta \cdot \mathcal{L}_{\text{SC}}. \tag{15}$$

Let $\alpha_{ij}$ denote the importance score of edge $e_{ij}$. The gradient w.r.t. $\alpha_{ij}$ is computed as:

$$\frac{\partial \mathcal{L}}{\partial \alpha_{ij}} \approx \underbrace{\frac{\partial \mathcal{L}_{\text{CE}}}{\partial \alpha_{ij}}}_{\text{classification}} + \beta \cdot \underbrace{\frac{\partial R(G_s)}{\partial \alpha_{ij}}}_{\text{conciseness}} + \eta \cdot \underbrace{\frac{\partial \mathcal{L}_{\text{SC}}}{\partial \alpha_{ij}}}_{\text{stability}}. \tag{16}$$

The three gradient components play different roles: the first term pushes important edges upward (toward 1) because they are needed for prediction; the second term depends on the specific form of CR: SMGNN encourages sparsity ($\alpha_{ij} \to 0$), whereas GSAT encourages independence ($\alpha_{ij} \to 0.5$); the third term stabilizes scores by pulling them toward near-fixed levels.

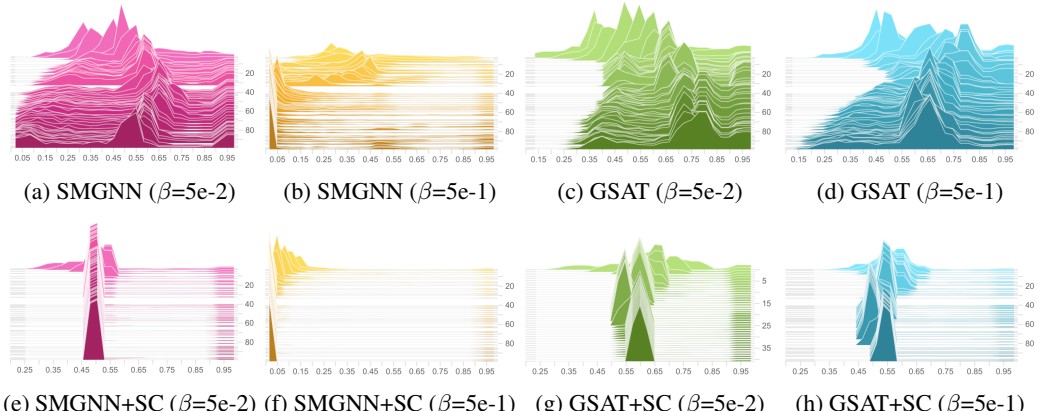

(a) SMGNN ($\beta$=5e-2)  (b) SMGNN ($\beta$=5e-1)  (c) GSAT ($\beta$=5e-2)  (d) GSAT ($\beta$=5e-1)

(e) SMGNN+SC ($\beta$=5e-2) (f) SMGNN+SC ($\beta$=5e-1) (g) GSAT+SC ($\beta$=5e-2)  (h) GSAT+SC ($\beta$=5e-1)

Figure 3: Distributions of edge scores assigned to unimportant edges (defined by the dataset ground truth) on BA-2MOTIFS dataset using SMGNN and GSAT. Each subplot shows the distribution of edge scores for unimportant edges over training epochs (vertical axis).

Table 1: Experimental results on SMGNN and GSAT. **Bold** numbers indicate improvements over the raw baselines. Best, and second-best results are highlighted in different colors.

| Method | BA-2MOTIFS | | 3MR | | BENZENE | | MUTAGENICITY | |
|---|---|---|---|---|---|---|---|---|
| | ↓ SHD (%) | ↑ AUC (%) | ↓ SHD (%) | ↑ AUC (%) | ↓ SHD (%) | ↑ AUC (%) | ↓ SHD (%) | ↑ AUC (%) |
| SMGNN | 10.44±4.41 | 99.32±0.36 | 9.44±2.24 | 97.00±0.77 | 16.06±6.57 | 84.38±2.71 | 12.65±3.18 | 98.12±0.39 |
| SMGNN+EE | **4.99±1.48** | **99.59±0.16** | **5.20±0.72** | **98.25±0.21** | **8.55±2.79** | **91.38±0.96** | **6.21±1.20** | **98.96±0.16** |
| SMGNN+SC | **3.48±2.20** | **99.87±0.05** | **5.47±3.39** | **98.39±0.88** | **7.19±2.62** | **90.07±1.56** | **2.51±1.02** | **98.30±0.54** |
| SMGNN+SC+EE | **1.52±1.03** | **99.90±0.02** | **3.05±1.22** | **99.25±0.02** | **4.14±0.82** | **92.20±0.43** | **1.44±0.38** | **98.96±0.17** |
| GSAT | 4.58±1.81 | 98.44±0.61 | 12.35±3.55 | 98.38±0.31 | 6.93±2.75 | 90.66±0.89 | 10.08±4.58 | 99.01±0.31 |
| GSAT+EE | **2.10±0.67** | **98.80±0.16** | **5.79±1.07** | **99.16±0.03** | **3.25±1.09** | **92.18±0.59** | **4.70±1.60** | **99.36±0.05** |
| GSAT+SC | **2.73±1.02** | **99.30±0.12** | **6.67±3.73** | **98.91±0.29** | **2.32±0.62** | **92.80±0.36** | **2.38±0.84** | **99.38±0.05** |
| GSAT+SC+EE | **1.19±0.52** | **99.35±0.03** | **4.00±0.38** | **99.39±0.01** | **1.14±0.26** | **93.53±0.11** | **1.06±0.42** | **99.41±0.01** |
| | ↓ FID (%) | ↑ ACC (%) | ↓ FID (%) | ↑ ACC (%) | ↓ FID (%) | ↑ ACC (%) | ↓ FID (%) | ↑ ACC (%) |
| SMGNN | 0.50±0.92 | 95.50±12.52 | 1.90±1.01 | 97.30±1.31 | 2.07±0.62 | 91.12±0.62 | 1.72±1.09 | 89.53±0.99 |
| SMGNN+EE | – | **100.00±0.00** | – | **98.25±0.21** | – | **92.16±0.24** | – | **90.36±0.70** |
| SMGNN+SC | **0.00±0.00** | **99.80±0.60** | **0.00±0.00** | **99.65±0.00** | **0.97±0.59** | **92.57±0.44** | **0.61±0.54** | **91.18±1.07** |
| SMGNN+SC+EE | – | **100.00±0.00** | – | **99.65±0.00** | – | **93.24±0.19** | – | **92.11±0.45** |
| GSAT | 0.00±0.00 | 100.00±0.00 | 0.90±0.28 | 98.55±0.80 | 1.80±0.84 | 91.48±0.87 | 1.11±0.61 | 92.43±1.00 |
| GSAT+EE | – | **100.00±0.00** | – | **99.15±0.24** | – | **92.17±0.28** | – | **93.00±0.44** |
| GSAT+SC | **0.00±0.00** | **100.00±0.00** | **0.31±0.29** | **99.55±0.16** | **0.74±0.14** | **92.31±0.32** | **0.17±0.23** | **93.48±0.48** |
| GSAT+SC+EE | – | **100.00±0.00** | – | **99.65±0.00** | – | **92.61±0.18** | – | **93.76±0.37** |

As a result, when $\beta$ is weak, CR barely influences edge scores. Important edges are driven to 1 by the classification loss, while unimportant edges can freely occupy any feasible near-fixed level under SC; When $\beta$ is moderately strong, CR actively suppresses unimportant edges: under SMGNN they are pushed toward 0, while under GSAT they converge toward 0.5. Important edges remain near 1, as the classification term dominates their gradients. This regime achieves the best explanation quality, since CR and SC jointly drive unimportant edges toward low and stable importance scores (Figure 3). When $\beta$ is strong, CR overwhelms classification: in SMGNN both important and unimportant edges collapse near 0, while in GSAT they collapse near 0.5. Consequently, explanation quality degrades, as important edges can no longer be distinguished from unimportant ones (Figure 14 in Section D.1).

## 4 EXPERIMENTS

**Datasets.** Following prior work (Tai et al., 2025), we evaluate on four benchmark datasets: the synthetic dataset BA-2MOTIFS (Luo et al., 2020) and three real-world molecular datasets—3MR (Rao et al., 2022), BENZENE Morris et al. (2020), and MUTAGENICITY (Morris et al., 2020).

**Baselines.** We implement SC on four representative SI-GNN backbones mentioned in Section 2.1. In addition, we compare against Explanation Ensemble (EE) (Tai et al., 2025), a post-hoc strategy designed to mitigate redundancy and improve explanation quality.

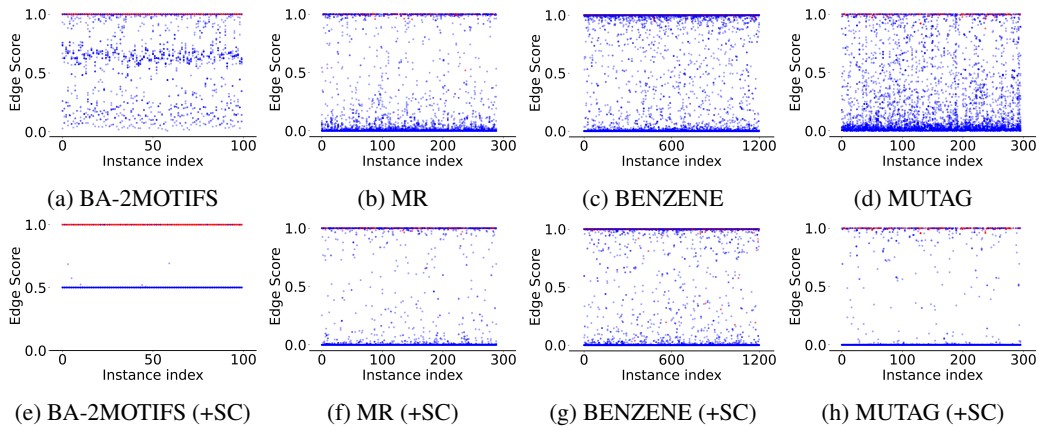

Figure 4: Distributions of edge scores assigned to important and unimportant edges (defined by the dataset ground truth) across datasets using SMGNN. The first row shows vanilla SI-GNNs, while the second row shows models trained with SC. Results for GSAT are provided in Figure 15.

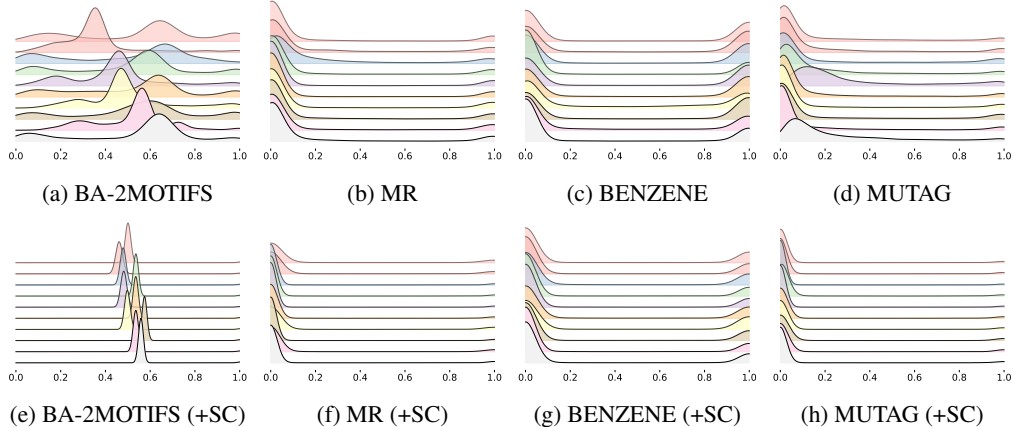

Figure 5: Unimportant edge weights across 10 SMGNN runs (GSAT results in Figure 16).

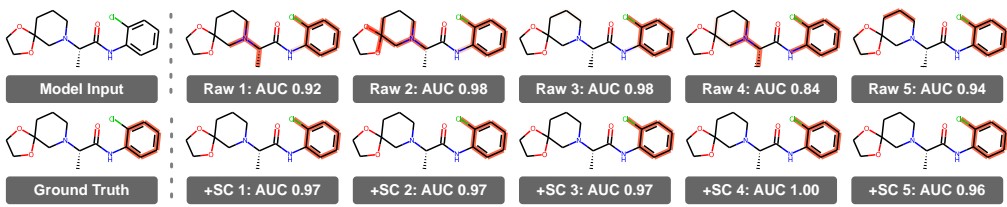

Figure 6: Explanations generated by five independently trained models (SMGNN vs. SMGNN+SC).

**Metrics.** We adopt four widely used metrics. Soft Structural Hamming Distance (SHD) measures explanation consistency, ROC-AUC (AUC) quantifies explanation accuracy, classification Accuracy (ACC) reflects informativeness for downstream prediction, and Fidelity- (FID) evaluates faithfulness. Further details about experimental settings are provided in Section C.

## 4.1 EFFECTIVENESS ON CR-BASED SI-GNNS

As discussed in Section 3.3, the effectiveness of SC depends on an appropriate strength of CR. We therefore first evaluate SC on two representative SI-GNNs with CR, SMGNN and GSAT. The results are summarized in Table 1. We draw three key observations: **(i) SC improves explanation quality**

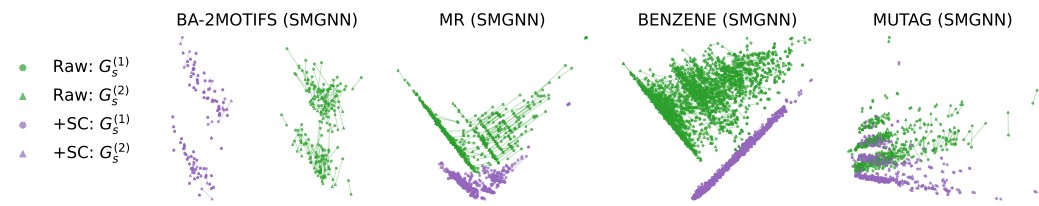

Figure 7: PCA visualization of explanation embeddings. Each line connects two successive representations of the same instance; shorter lines indicate better stability (GSAT results in Figure 17).

Table 2: Pairwise similarity between embeddings of $G_s^{(1)}$ and $G_s^{(2)}$ across 10 runs. Higher cosine (COS) similarity and lower L1 distance indicate better self-consistency.

| Method | BA-2MOTIFS | | MR | | BENZENE | | MUTAGENICITY | |
|---|---|---|---|---|---|---|---|---|
| | ↑ COS (%) | ↓ L1 (%) | ↑ COS (%) | ↓ L1 (%) | ↑ COS (%) | ↓ L1 (%) | ↑ COS (%) | ↓ L1 (%) |
| SMGNN | 99.68±0.38 | 16.87±8.30 | 95.81±6.90 | 51.01±32.81 | 99.11±1.63 | 33.23±30.76 | 98.79±2.56 | 48.18±45.07 |
| SMGNN+SC | **99.98±0.07** | **1.84±5.69** | **99.92±0.52** | **2.83±8.84** | **99.80±1.26** | **7.96±18.82** | **99.96±0.14** | **5.93±15.41** |
| GSAT | **98.87±1.56** | 23.82±12.05 | 95.57±10.52 | 12.35±12.99 | 99.15±1.63 | 29.99±17.60 | 99.66±0.65 | 19.84±16.43 |
| GSAT+SC | 98.71±2.18 | **17.09±14.31** | **98.65±4.92** | **6.30±9.94** | **99.66±1.23** | **9.59±11.68** | **99.94±0.19** | **5.71±6.79** |

Table 3: Experimental results on GAT and CAL. **Bold** numbers indicate improvements over the raw baselines. Best and second-best results are highlighted in different colors.

| Method | BA-2MOTIFS | | MR | | BENZENE | | MUTAGENICITY | |
|---|---|---|---|---|---|---|---|---|
| | ↓ SHD (%) | ↑ AUC (%) | ↓ SHD (%) | ↑ AUC (%) | ↓ SHD (%) | ↑ AUC (%) | ↓ SHD (%) | ↑ AUC (%) |
| GAT | 10.41±4.37 | 99.31±0.35 | 14.04±2.05 | 97.25±0.67 | 14.28±5.76 | 83.51±2.24 | 11.85±7.47 | 91.71±6.00 |
| GAT+SC | 16.63±15.21 | 99.14±2.15 | **8.30±3.31** | **97.66±1.00** | 19.43±9.54 | **85.13±5.09** | **0.04±0.05** | 81.79±12.31 |
| GAT+CR+SC | **3.48±2.20** | **99.87±0.05** | **5.47±3.39** | **98.39±0.88** | **7.19±2.62** | **90.07±1.56** | **2.51±1.02** | **98.30±0.54** |
| CAL | 11.40±4.04 | 98.64±1.33 | 14.51±4.33 | 96.25±1.59 | 21.25±8.91 | 77.64±3.05 | 21.82±11.15 | 96.39±1.39 |
| CAL+SC | 17.54±15.31 | **98.80±2.41** | **9.08±3.54** | **97.23±0.49** | **11.32±7.90** | **88.74±3.06** | **0.01±0.01** | **96.52±2.07** |
| CAL+CR+SC | **1.90±1.54** | **98.90±2.44** | **7.92±2.40** | **97.64±0.76** | **6.25±2.07** | **89.87±0.98** | **2.78±1.52** | **97.93±0.46** |
| | ↓ FID (%) | ↑ ACC (%) | ↓ FID (%) | ↑ ACC (%) | ↓ FID (%) | ↑ ACC (%) | ↓ FID (%) | ↑ ACC (%) |
| GAT | 2.50±7.50 | 97.10±8.70 | 2.63±1.93 | 96.54±1.46 | 2.06±0.65 | 91.60±0.66 | 0.37±0.53 | 92.97±0.78 |
| GAT+SC | **0.00±0.00** | **100.00±0.00** | **0.10±0.22** | **99.55±0.16** | **0.82±0.61** | **92.46±0.32** | **0.00±0.00** | **93.65±0.77** |
| GAT+CR+SC | **0.00±0.00** | **100.00±0.00** | **0.00±0.00** | **99.65±0.00** | **0.97±0.59** | **92.57±0.44** | 0.61±0.54 | 91.18±1.07 |
| CAL | 17.10±17.21 | 91.90±11.68 | 8.17±2.52 | 94.22±2.11 | 4.97±3.44 | 84.31±5.66 | 2.06±1.14 | 91.22±1.37 |
| CAL+SC | **0.00±0.00** | **100.00±0.00** | **1.56±0.76** | **97.82±0.97** | **2.45±0.92** | **91.77±0.45** | **0.03±0.10** | **93.11±0.70** |
| CAL+CR+SC | **0.00±0.00** | **100.00±0.00** | **5.02±5.86** | **98.27±0.93** | **3.11±1.11** | **91.49±0.54** | 4.22±1.20 | 90.88±0.71 |

**in all cases.** Compared to the vanilla SI-GNN, SC consistently enhances explanation consistency (SHD), accuracy (AUC), faithfulness (FID), and informativeness (ACC). **(ii) SC outperforms EE in most cases.** Despite EE's ability to mitigate redundancy, SC achieves better performance while being more efficient (∼5x faster) and more general (compatible with all standard evaluation metrics). This highlights SC as an appealing alternative pathway for addressing redundancy. **(iii) SC and EE are complementary.** When combined, SC+EE yields further gains over either strategy alone, suggesting that SC and EE tackle redundancy from different, reinforcing perspectives.

To better understand how SC improves explanation quality, we complement the main results with visualizations and case studies. Figure 4 shows the distributions of edge scores for important and unimportant edges across four datasets with $\beta = 0.05$. Consistent with our analysis in Section 3.2, SC drives important edges toward 1 while pushing unimportant edges toward other low, near-fixed levels, resulting in more stable assignments. Figure 5 shows that the collective allocation induced by SC is stable: under the same conciseness regularization, score assignments remain similar across runs, yielding consistent explanations. A case study on the BENZENE dataset (Figure 6) further illustrates that, after SC training, the explainer assigns importance scores more responsibly. Complementary results for GSAT are provided in Section D.1.

Since SC explicitly reduces discrepancies between successive explanations, it naturally lowers FID: consistent explanations yield consistent representations and predictions. Figure 7 shows PCA plots where SC produces visibly shorter connections between successive runs, and Table 2 confirms with

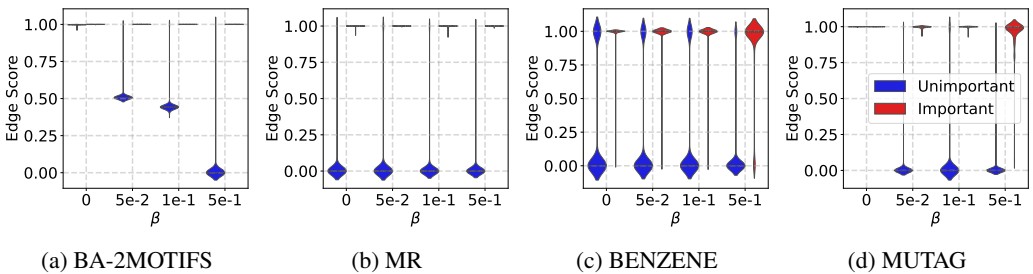

Figure 8: Edge scores under different values of $\beta$ using SMGNN (GSAT results in Figure 18).

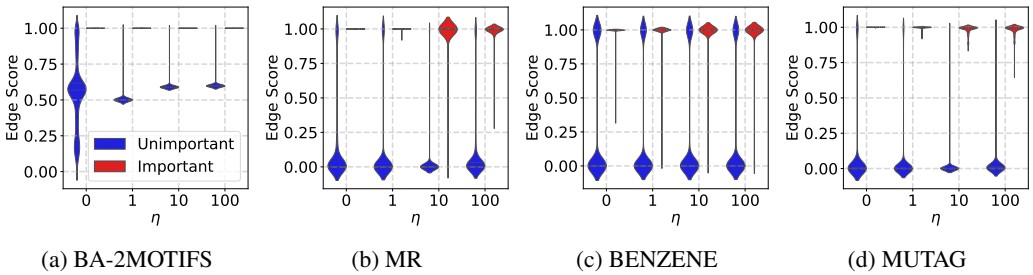

Figure 9: Edge scores under different values of $\eta$ using SMGNN (GSAT results in Figure 19).

higher cosine similarity and lower L1 distance. To further validate effectiveness and rule out confounds such as extended training time (pretraining the GNN encoder and fine-tuning the explainer and classifier), we include additional experiments in Section D.2.

## 4.2 EFFECTIVENESS ON NON-CR-BASED SI-GNNS

As discussed in Section 2.2, GAT and CAL lack explicit conciseness constraints. In this setting, applying SC alone may cause unimportant edges to drift toward high scores (e.g., Figures 12e and 12h and Figure 13h in Section D.1), which instead degrades explanation quality. Table 3 confirms this: for instance, GAT+SC's AUC drops on BA-2MOTIFS and MUTAGENICITY. When CR is added, SC improves explanation quality across all datasets in 28 out of 32 cases. Importantly, without CR the degradation can be severe (e.g., SHD increases by 6% and AUC drops by 10%), whereas with CR even the few unfavorable cases show only marginal gaps (1–2%). This contrast highlights that CR provides a stable foundation on which SC can consistently enhance explanation quality.

## 4.3 PARAMETER SENSITIVITY

Figures 8 and 9 analyze the effect of the two hyperparameters $\beta$ and $\eta$. Consistent with our analysis in Section 3.3, $\beta$ can influence the selection of near-fixed levels. For example, in Figure 8a, when $\beta = 0$, even unimportant edges may be pushed toward 1; as $\beta$ increases, their scores shift toward 0, leading to a clearer separation between important and unimportant edges. In practice, $\beta$ is selected by maximizing validation accuracy and adjusted through heuristic checks—e.g., whether importance scores are sufficiently separated to yield informative explanations. As a result, $\beta$ typically lies in a suitable range, and our strategy can be directly applied without additional tuning.

Compared to $\beta$, our method is less sensitive to $\eta$. As $\eta$ increases, unimportant edges that would otherwise receive high scores are gradually suppressed toward lower values, while important ones remain unaffected; even very large $\eta$ yield relatively stable results. This may be due to freezing the encoder, which prevents SC from perturbing the representation space. Moreover, since SC primarily targets redundant edges (Figure 1), its optimization rarely conflicts with the main training objective. We defer a more detailed discussion to Section E. Notably, $\eta$ is the only additional parameter introduced by our strategy, yet its low sensitivity makes the approach readily applicable in practice.

## 5 CONCLUSION

In this work, we discussed the fundamental mismatch between the training objectives of SI-GNNs and the faithfulness criteria by which they are evaluated. We showed that faithfulness is inherently tied to explanation self-consistency, and that enforcing self-consistency during training can serve as a direct and principled way to optimize faithfulness. Our analysis revealed that self-inconsistency arises mainly on unimportant edges, linking it to redundancy and motivating a simple yet effective solution SC. Extensive experiments across diverse benchmarks demonstrated that SC consistently improves explanation consistency, accuracy, faithfulness, and informativeness, while remaining easy to apply in practice. In future work, we plan to extend SC from graph benchmarks to AI4Science domains, where trustworthy explanations are crucial for advancing scientific discovery.

## ACKNOWLEDGEMENT

We thank Zeng Wang for her constructive discussions and valuable feedback. We also thank Yaqian Liu for her help with proofreading the manuscript and double-checking the experimental results. This work was supported in part by the National Natural Science Foundation of China (Grant No. 62176043, supporting Ting Zhong; and Grant No. 62572097, supporting Fan Zhou).

## ETHICS STATEMENT

This work uses only publicly available benchmark datasets for graph explanation tasks. No human subjects, personally identifiable information, or sensitive data are involved. Our work aims to improve explanation quality, which we believe has positive implications for transparency and trustworthiness in AI systems.

## REPRODUCIBILITY STATEMENT

We have made every effort to ensure reproducibility of our results. All datasets used in this work are publicly available benchmark datasets for graph explanation tasks. The details of training procedures, model architectures, and hyperparameter settings are provided in Sections B and C. Each experimental result is averaged over 10 independent runs to ensure reliability. An implementation of our method, along with training scripts and pre-trained model checkpoints, is provided on Github.

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

APPENDIX

## A  RELATED WORK

Self-Interpretable GNNs (SI-GNNs) are designed to generate explanations as an intrinsic part of their prediction process (Ji et al., 2025). By coupling interpretation with prediction, they aim to produce explanations that are more faithful to the model's decision-making than post-hoc approaches (Ying et al., 2019; Luo et al., 2020; Pope et al., 2019; Yuan et al., 2021; Zhang et al., 2022; Chen et al., 2024; Luo et al., 2024). A typical design pattern involves two stages: (i) selecting a critical graph subset, often through soft or hard masks over edges or nodes, to serve as the explanation; and (ii) making the final prediction based solely on this explanatory subset.

Existing approaches differ mainly in how they extract the explanatory subgraph (Tai et al., 2025). Attention-based approaches (Velickovic et al., 2018; Knyazev et al., 2019; Lu & Li, 2020) directly assign importance scores via attention mechanisms without explicit constraints on conciseness. Causal-learning methods (Wu et al., 2022b; Sui et al., 2022) employ disentanglement and interventions (Pearl, 2014) to isolate causal factors. Size-constrained methods (Lin et al., 2020; Luo et al., 2024; Azzolin et al., 2025b) encourage concise explanations by penalizing subset size or sparsity. Mutual information (MI)-constrained methods (Yu et al., 2021; 2022; Miao et al., 2022; Seo et al., 2023) minimize the MI between the explanatory subset and the full graph to enforce conciseness.

Our work is complementary to these directions: instead of introducing new subset selection strategies, we focus on enforcing self-consistency during training. This provides a simple, model-agnostic way to optimize faithfulness and enhance explanation quality across different SI-GNNs directly.

Beyond the SI-GNNs discussed above, two recent studies provide relevant perspectives for our work. Azzolin et al. (2025a) questioned the reliability of faithfulness itself, showing that good faithfulness scores do not necessarily imply high-quality explanations. Their observation directly inspired our second research question—whether explicitly optimizing faithfulness can genuinely improve explanation quality—and motivated us to adopt a broader evaluation that considers explanation consistency, accuracy, faithfulness, and informativeness together. Tai et al. (2025) investigated explanation inconsistency across repeated training runs of the same SI-GNN and traced it to explanation redundancy, i.e., the tendency of explainers to irresponsibly assign high importance to unimportant edges when budget allows. Our study corroborates their findings and further shows that redundancy also underlies a related form of instability—self-inconsistency within a single SI-GNN. Inspired by these insights, we hypothesized that mitigating self-inconsistency should also mitigate redundancy, and thereby improve explanation quality—a link already established in Tai et al. (2025).

Our experiments confirm this hypothesis: under certain conditions, self-consistency training indeed improves explanation quality. More broadly, our work enriches the characterization of redundancy by revealing its manifestation as self-inconsistency within a single SI-GNN, and further provides a new perspective for mitigation through a simple, efficient, and model-agnostic training strategy.

## B  SI-GNN PIPELINES

The generic pipeline of SI-GNNs can be described as follows. A GNN encoder first computes node embeddings:

$$\mathbf{V} = \text{GNN}(G). \tag{17}$$

An explainer module (typically an MLP) then estimates edge importance scores from the embeddings of incident nodes:

$$w_{ij} = \text{MLP}([\mathbf{v}_i; \mathbf{v}_j]), \quad \alpha_{ij} = \sigma(w_{ij}), \tag{18}$$

where $[\mathbf{v}_i; \mathbf{v}_j]$ denotes concatenation and $\sigma$ is the sigmoid. Each edge $e_{ij}$ is included in the explanatory subgraph $G_s$ with probability $\alpha_{ij}$, yielding the factorized distribution:

$$\mathbb{P}(G_s) = \prod_{e_{ij} \in \mathcal{E}} \text{Bern}(\alpha_{ij}). \tag{19}$$

To enable differentiable sampling during training, we follow Miao et al. (2022); Luo et al. (2024) and employ the Gumbel–Sigmoid trick (Jang et al., 2017):

$$\mathbb{P}(e_{ij}) \leftarrow \sigma((\ln \epsilon - \ln(1 - \epsilon) + w_{ij})/\tau), \tag{20}$$

with $\epsilon \sim \mathrm{Uniform}(0, 1)$ and temperature $\tau$. At inference, the score $\alpha_{ij}$ is used directly.

The sampled subgraph $G_s$ is processed by the same encoder to obtain a graph-level embedding:

$$\mathbf{z}_s = \mathrm{POOL}(\mathrm{GNN}(G_s)), \tag{21}$$

where POOL is a permutation-invariant readout (e.g., sum, mean, or max). A classifier MLP finally maps $\mathbf{z}_s$ to the prediction:

$$\hat{Y} = \mathrm{MLP}(\mathbf{z}_s). \tag{22}$$

The training objectives follow Section 2.1. Note that CAL differs from the other three SI-GNNs in how it handles the explanatory subset. While Equation (21) describes the standard formulation where $G_s$ is reused as input to the encoder, CAL does not reuse $G_s$ in this way. More specifically, CAL has three classifiers corresponding to the three terms in Equation (3): (1) the first classifier takes $G_s$ as input; (2) the second classifier takes $\bar{G}_s$ as input; (3) the third classifier takes $G_s \cup \bar{G}'_s$ as input. During training, CAL applies intervention to $\bar{G}_s$, but no intervention is applied at inference time. Therefore, the third classifier actually receives the full graph $G$ at inference, and its output is used as CAL's final prediction.

**Remark on SC Training.** While the generic SI-GNN pipeline employs the Gumbel–Sigmoid trick during training to enable differentiable sampling, we make a slight modification for SC training. Specifically, when constructing the paired subgraphs $G_s^{(1)}$ and $G_s^{(2)}$ used in the SC loss, we bypass the stochastic sampling step and instead use the deterministic edge scores to form $G_s^{(1)}$ and $G_s^{(2)}$. This design avoids the additional randomness introduced by Gumbel noise, which would otherwise confound the SC alignment. At inference, we use $\alpha_{ij}$ as the final importance scores, the same as prior work (Miao et al., 2022).

## C EXPERIMENTAL SETTINGS

**Baselines.** We evaluate our method on four representative SI-GNNs, each reflecting a different design principle: GAT (Velickovic et al., 2018), SMGNN (Lin et al., 2020), CAL (Sui et al., 2022), and GSAT (Miao et al., 2022). All baselines are re-implemented under a unified training framework to ensure comparability (Tai et al., 2025). Specifically, GAT is trained with the standard classification loss. SMGNN combines classification with a sparsity regularizer, omitting the entropy term since Gumbel sampling already produces near-binary masks (Tai et al., 2025). CAL follows its original 3-classifier formulation, but for stability we replace its 1-layer classifiers with 3-layer MLPs, applying this change consistently across all baselines.

In addition, we compare against Explanation Ensemble (EE), a post-hoc strategy designed to mitigate redundancy and improve explanation quality (Tai et al., 2025). For each SI-GNN type, we train 10 models with different random seeds. EE is then constructed within each type by ensembling five models. This choice is dictated by the SHD metric: computing SHD requires two independent ensembles, and using more than five models would force reuse across ensembles, undermining the validity of the comparison.

**Metrics.** We adopt four metrics, each probing a distinct aspect of explanation quality.

*Consistency (edge-level).* To measure explanation stability, we follow Tai et al. (2025) and compute the Structural Hamming Distance (SHD) (Tsamardinos et al., 2006) across runs with different random seeds:

$$\mathrm{SHD}(e_{ij}) = \frac{2}{N(N-1)} \sum_{1 \leq p < q \leq N} \left| \alpha_{ij}^{(p)} - \alpha_{ij}^{(q)} \right|, \tag{23}$$

where $N$ is the number of runs. Unlike prior work that thresholds edge weights (e.g., at 0.5), we compute SHD directly on continuous scores, avoiding arbitrary cutoffs that can strongly bias results.

*Accuracy (instance-level).* Following Wu et al. (2022b); Miao et al. (2022); Rao et al. (2022), we use ROC-AUC (AUC) to evaluate whether annotated ground-truth edges receive higher importance than irrelevant ones. AUC is particularly useful for detecting redundancy, but its reliability depends on the reliability of the ground-truth annotations. As verified in Tai et al. (2025), the datasets we use align well with the model's decision rationale, ensuring valid evaluations.

*Faithfulness (instance-level).* Faithfulness evaluates whether the selected subgraph $G_s$ preserves the model's decision (Yuan et al., 2022). We adopt Fidelity$^-$ (FID$^-$), which we simply denote as FID in the main text, defined as:

$$\text{FID}^-(G, G_s) = 1 - \mathbb{1}\big(c(f(G_s)) = c(f(G))\big), \tag{24}$$

where $c(\cdot)$ is the predicted class. An alternative is Fidelity$^+$ (FID$^+$), which checks whether removing $G_s$ changes the prediction:

$$\text{FID}^+(G, G_s) = 1 - \mathbb{1}\big(c(f(G \setminus G_s)) = c(f(G))\big), \tag{25}$$

However, for SI-GNNs, FID$^+$ confounds explanation quality with distribution-shift generalization, since $G \setminus G_s$ is never used to train the GNN encoder. Therefore, we report FID$^-$ in the main text, while including FID$^+$ results in Section D.2 for completeness.

*Informativeness (instance-level).* Finally, we report classification Accuracy (ACC) (Wu et al., 2022b; Miao et al., 2022; Amara et al., 2022). While not a direct metric of explanation quality, ACC serves as a sanity check: if the explanations are informative, predictive performance should be preserved when restricted to explanatory subsets.

**Implementations.** We follow the setup described in prior work (Tai et al., 2025) and adopt a 2-layer GIN (Xu et al., 2019) encoder (hidden sizes $64, 64$) with a dropout rate of $0.3$. The explainer is a 3-layer MLP (hidden sizes $256, 64, 1$) with a dropout rate of $0.5$ for predicting edge weights, and the classifier is a 3-layer MLP (hidden sizes $64, 64, 1$) for final predictions. Hyperparameters are tuned on validation sets: learning rate from $\{0.01, 0.005, 0.001, 0.0005, 0.0001\}$, and regularization coefficients (e.g., $\beta, \gamma$) from $\{0.01, 0.05, 0.1, 0.5, 1, 5, 10, 50, 100\}$, guided primarily by classification accuracy with additional heuristics on explanation conciseness. All experiments are repeated 10 times with different random seeds to ensure reliability. Implementations are based on PyTorch and training is performed with the Adam optimizer (Kingma & Ba, 2015).

Because our method follows a two-stage training scheme, we directly use the checkpoints[1] released by Tai et al. (2025) as the outcome of the first stage. On top of these, we fine-tune the explainer and classifier for 100 epochs while keeping the GNN encoder frozen. All hyperparameters are kept identical to those in Tai et al. (2025), with the only additional hyperparameter being the SC coefficient $\eta$, which we set to 1 for all experiments. For BA-2MOTIFS, we found that the $\beta$ setting reported by Tai et al. (2025) (5e-2) yields relatively weak conciseness constraints; we therefore also consider a stronger $\beta$ (5e-1). For GAT, the +CR variant corresponds to SMGNN, while CAL+CR follows the same implementation as SMGNN.

## D  MORE EXPERIMENTS

For completeness, we provide additional experimental results in this section. We divide the presentation into two parts:

- **Complementary results.** These are experiments already discussed in the main text, where we reported only a subset of results due to space limitations.
- **New experiments.** Beyond the main text, we conduct further evaluations to give a more comprehensive assessment of our method. These additional experiments serve to validate robustness, generality, and provide deeper insights into the behavior of SC.

### D.1  COMPLEMENTARY RESULTS

Figure 10–Figure 13 present complementary results on all four classes of SI-GNNs, where we explicitly track the evolution of edge scores assigned to unimportant edges during training with SC. For

---

[1] `https://github.com/ICDM-UESTC/TrustworthyExplanation/releases/tag/archive`

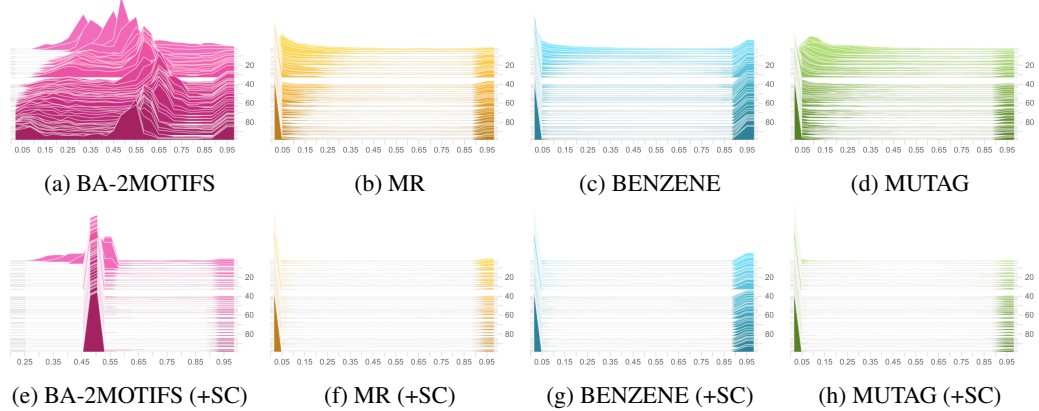

Figure 10: Distributions of edge scores assigned to unimportant edges (defined by the dataset ground truth) across datasets using SMGNN. The first row shows vanilla SI-GNNs, while the second row shows models trained with SC. Each subplot shows the distribution of edge scores for unimportant edges over training epochs (vertical axis).

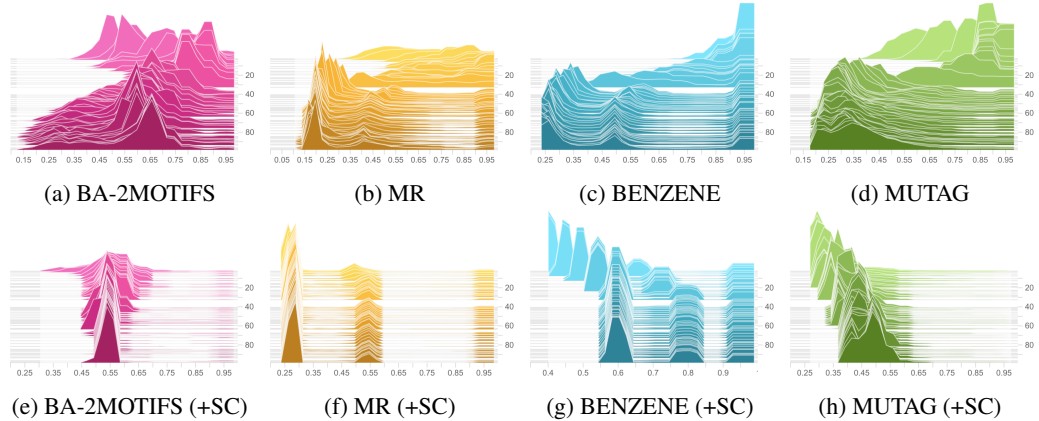

Figure 11: Distributions of edge scores assigned to unimportant edges across datasets using GSAT. Plotting convention follows Figure 10.

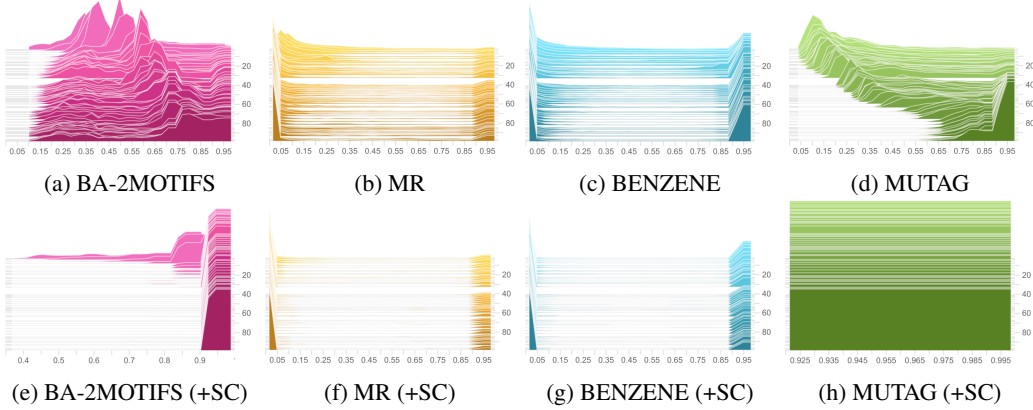

Figure 12: Distributions of edge scores assigned to unimportant edges across datasets using GAT. Plotting convention follows Figure 10.

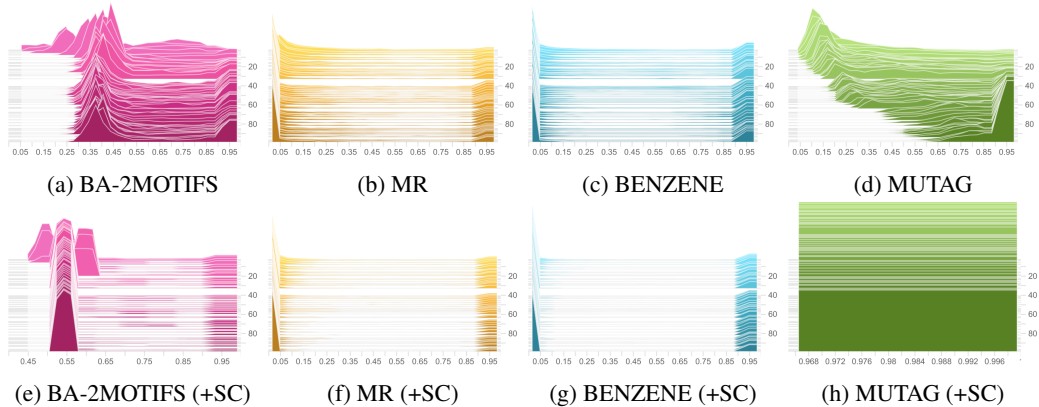

Figure 13: Distributions of edge scores assigned to unimportant edges across datasets using CAL. Plotting convention follows Figure 10.

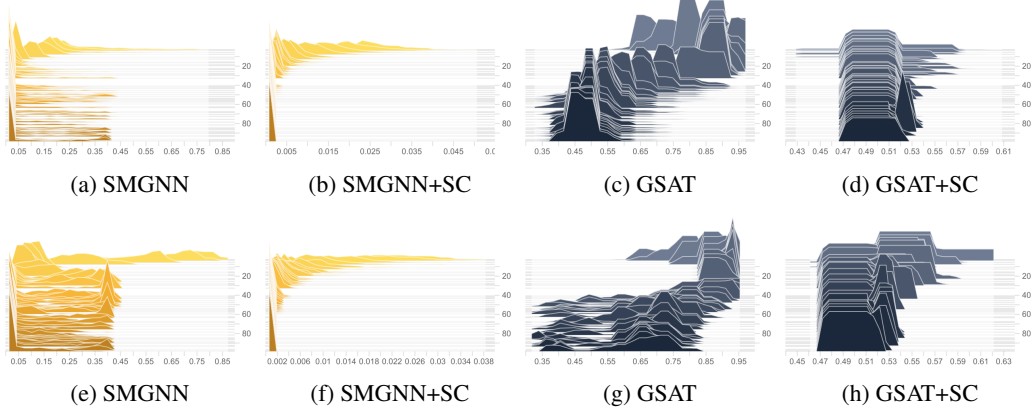

Figure 14: Impact of excessively strong CR regularization on BA-2MOTIFS with SMGNN and GSAT. The top row shows the distributions of edge scores for unimportant edges (UNIMP), and the bottom row shows those for important edges (IMP). When $\beta$ is overly large, CR collapses both important and unimportant edges (toward $0$ in SMGNN and $0.5$ in GSAT).

SMGNN and GSAT, we observe that unimportant edges converge toward stable low-score near fixed points. In contrast, for GAT and CAL, which lack CR regularization, applying SC alone may instead push uninformative edges toward arbitrarily high scores (e.g., Figures 12e and 12h and Figure 13h).

Figure 14 illustrates the impact of excessively strong CR regularization. When $\beta$ is too large (5e0 for SMGNN and 5e1 for GSAT), CR dominates the training dynamics: in SMGNN both important and unimportant edges collapse near $0$, while in GSAT they collapse around $0.5$. Without SC, the distributional shape of edge scores is already distorted; consequently, applying SC merely forces these distorted scores to concentrate further. This highlights that SC is effective only when the CR strength is set within a reasonable range, where the underlying separation between important and unimportant edges is still preserved.

Figure 15 extends the scatter plot analysis of Figure 4 to GSAT. We observe a similar effect as in SMGNN: with SC, important edges are consistently driven toward high scores (near $1$), while unimportant edges converge to lower, near-fixed levels. This separation reduces variance across runs and yields more stable and interpretable assignments, demonstrating that the effect of SC is not limited to size-constrained regularization but generalizes to MI-constrained models like GSAT.

Figure 16 presents the ridge plots for GSAT, complementing the SMGNN results in the main text (Figure 5). We again find that, without SC, the importance scores of uninformative edges fluctuate substantially across runs, reflecting the instability (irresponsibility) of the explainer. In con-

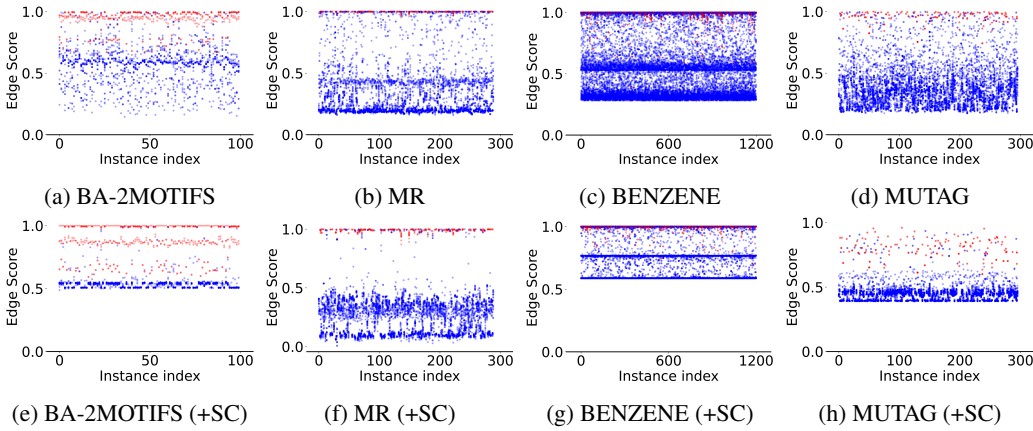

Figure 15: Distributions of edge scores assigned to important and unimportant edges across datasets using GSAT. Plotting convention follows Figure 4.

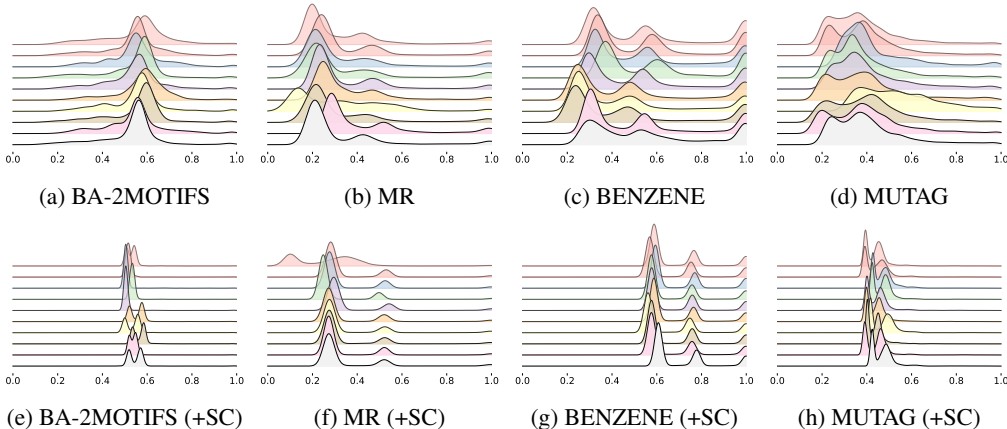

Figure 16: Unimportant edge weights across 10 GSAT runs.

trast, SC encourages these scores to concentrate around stable, low values, making their allocation consistent across different seeds. This confirms that the stabilizing effect of SC generalizes from size-constrained to MI-constrained SI-GNNs.

Figure 17 provides a complementary visualization for GSAT. As with the results reported in the main text, we observe that SC training substantially reduces the distance between successive explanation embeddings, leading to shorter connections in the PCA space. This result confirms that the stability benefit of SC generalizes beyond size-constrained SI-GNNs to MI-constrained SI-GNNs. Together with the quantitative similarity results in Table 2, these visualizations highlight that SC consistently enhances explanation faithfulness across different SI-GNNs.

Figures 18 and 19 complement the SMGNN results in the main text by showing the sensitivity of GSAT to the two hyperparameters $\beta$ and $\eta$. Consistent with our earlier analysis, $\beta$ governs the separation between important and unimportant edges by determining their selected near-fixed levels. Compared to $\beta$, $\eta$ exhibits greater stability, as it generally pushes unimportant edges toward low values while preserving high scores for important ones. These results confirm that our conclusions about parameter sensitivity generalize across both size-constrained and MI-constrained SI-GNNs.

Note that instead of evaluating SC on both GIN and GCN (Kipf & Welling, 2017), we report results using GIN only. Empirically, we observe that SC does not behave properly on GCN: when SC is applied, nearly all edges tend to receive high importance scores. In fact, even without SC, the original GCN explanations exhibit very weak discriminability—the absolute gap between important and unimportant edges is very small. We conjecture that this phenomenon is related to the aggregation

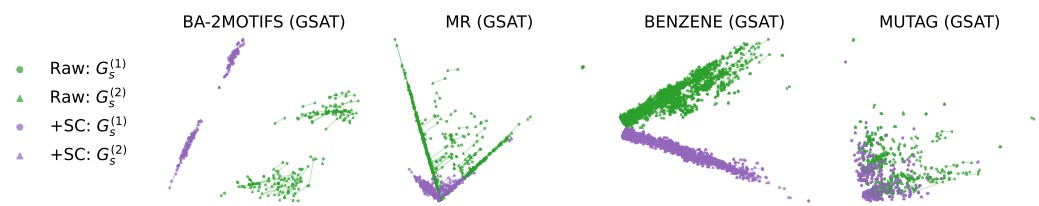

Figure 17: PCA visualization of explanation embeddings for GSAT. Each line connects two successive representations of the same instance; shorter lines indicate higher stability.

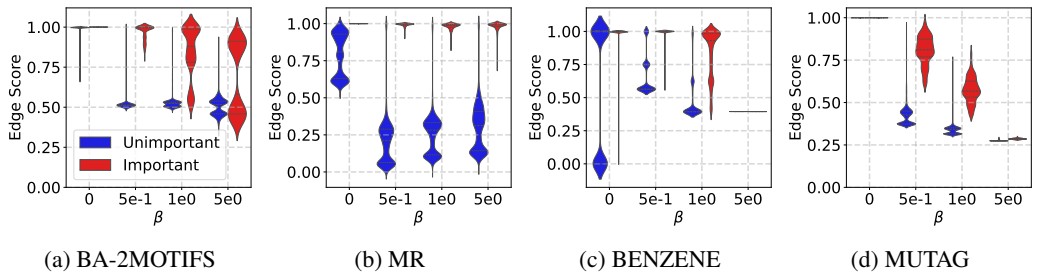

Figure 18: Edge scores under different values of $\beta$ using GSAT.

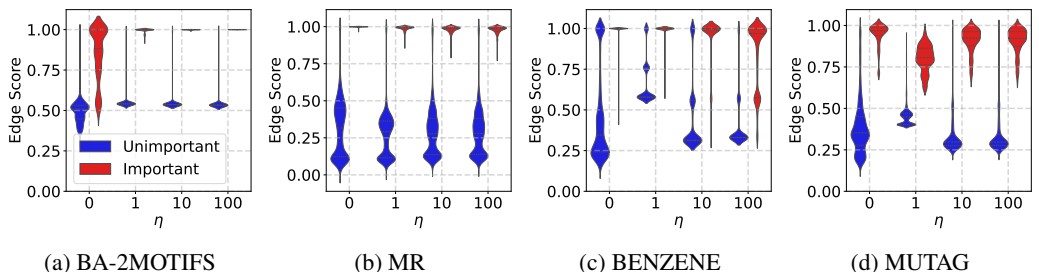

Figure 19: Edge scores under different values of $\eta$ using GSAT.

mechanism of GCN. After multiple rounds of message passing, node representations tend to become smoothed, reducing inter-node distinguishability and making it difficult for the explainer to extract effective structural signals. We further evaluated SC on GatedGCN (Bresson & Laurent, 2017) and GraphSAGE (Hamilton et al., 2017), whose architectural designs better preserve representational diversity across layers. In these settings, SC remains effective.

## D.2 NEW EXPERIMENTS

To further validate the robustness of our findings and address potential concerns about evaluation and training design, we conduct three complementary sets of additional experiments. First, we report results under Fidelity$^+$ (FID$^+$) to provide a more complete view of faithfulness evaluation. Second, we examine whether the observed improvements of SC could simply stem from extended training in our two-stage framework. Third, we analyze the necessity of freezing the encoder during SC fine-tuning by evaluating several alternative training configurations.

### D.2.1 FIDELITY EVALUATION

For completeness, we also report results under Fidelity$^+$ (FID$^+$), the other widely-used measure of faithfulness that evaluates whether removing the explanatory subgraph $G_s$ changes the model's prediction. Unlike FID$^-$, which directly measures whether $G_s$ alone suffices for the prediction, FID$^+$ inherently mixes explanation quality with the encoder's ability to generalize under distribution shift. Since SI-GNN encoders are never trained on $G \setminus G_s$, their representations on such inputs can

Table 4: Fidelity evaluation on four SI-GNNs. **Bold** indicates improvements over the raw baselines.

| Method | BA-2MOTIFS | | MR | | BENZENE | | MUTAGENICITY | |
|---|---|---|---|---|---|---|---|---|
| | ↓ FID⁻ (%) | ↑ FID⁺ (%) | ↓ FID⁻ (%) | ↑ FID⁺ (%) | ↓ FID⁻ (%) | ↑ FID⁺ (%) | ↓ FID⁻ (%) | ↑ FID⁺ (%) |
| SMGNN | 0.50±0.92 | 42.60±13.82 | 1.90±1.01 | 40.83±1.33 | 2.07±0.62 | 52.99±2.03 | 1.72±1.09 | 8.68±2.98 |
| SMGNN+SC | **0.00±0.00** | **52.60±3.29** | **0.00±0.00** | 39.20±0.62 | **0.97±0.59** | **55.35±0.88** | **0.61±0.54** | **40.24±1.94** |
| GSAT | **0.00±0.00** | 46.40±6.65 | 0.90±0.28 | 36.12±3.92 | 1.80±0.84 | 42.90±10.18 | 1.11±0.61 | 2.30±1.90 |
| GSAT+SC | **0.00±0.00** | **48.20±11.13** | **0.31±0.29** | **39.41±0.55** | **0.74±0.14** | 28.58±12.25 | **0.17±0.23** | 0.74±0.50 |
| GAT | 2.50±7.50 | 27.40±15.96 | 2.63±1.93 | 41.04±1.19 | 2.06±0.65 | 48.83±7.39 | 0.37±0.53 | 0.98±0.73 |
| GAT+SC | **0.00±0.00** | **36.30±24.17** | **0.21±0.35** | 38.62±1.04 | **0.80±0.37** | **54.96±1.20** | **0.00±0.00** | 0.00±0.00 |
| GAT+CR+SC | **0.00±0.00** | **52.60±3.29** | **0.00±0.00** | 39.20±0.62 | **0.97±0.59** | **55.35±0.88** | **0.61±0.54** | **40.24±1.94** |
| CAL | 17.10±17.21 | 36.60±8.82 | 8.17±2.52 | 33.67±5.86 | 4.97±3.44 | 41.15±8.54 | 2.06±1.14 | 5.00±8.18 |
| CAL+SC | **0.00±0.00** | **38.30±24.09** | **1.56±0.76** | **39.38±0.80** | **2.45±0.92** | **55.21±0.96** | **0.03±0.10** | 0.10±0.30 |
| CAL+CR+SC | **0.00±0.00** | **52.90±3.96** | **5.02±5.86** | **39.90±0.83** | **3.11±1.11** | **56.08±1.20** | 4.22±1.20 | **40.44±4.05** |

be unreliable. This confound often obscures the actual contribution of the explainer. As shown in Table 4, our method with SC consistently improves over baselines on FID⁻, but the advantage on FID⁺ is less clear. This outcome is expected: SC is explicitly designed to enforce self-consistency on $G_s$, which directly aligns with FID⁻ but not with FID⁺. However, poor encoder generalization on $G \setminus G_s$ can significantly influence FID⁺. Therefore, while we provide FID⁺ for completeness, our main text focuses on FID⁻ together with other complementary metrics (consistency, accuracy, informativeness) for a more reliable and comprehensive assessment of explanation quality.

### D.2.2   DOES SC SIMPLY BENEFIT FROM LONGER TRAINING?

Since our framework adopts a two-stage design—first training the GNN encoder and then fine-tuning the explainer and classifier with the encoder frozen—one might suspect that the observed improvements come simply from longer training rather than SC itself. To rule this out, we conduct an ablation where we fix the encoder and fine-tune only the explainer and classifier under two settings: $\eta = 0$ (two-stage training without SC) and $\eta = 1$ (two-stage training with SC).

As shown in Table 5, fine-tuning without SC does not consistently improve explanation quality; in some cases it even degrades performance (e.g., GSAT on BA-2MOTIFS, where AUC drops by 13%). This indicates that extended training alone is insufficient for achieving better explanations. While fixing the encoder and fine-tuning the explainer and classifier can sometimes provide modest gains, this is not due to additional training epochs (our experiments show models already converge well before 100 epochs), but more likely because fixing the encoder avoids feature drift from joint optimization and thereby stabilizes training. By contrast, once SC is introduced, explanation quality consistently improves across all datasets and metrics, outperforming not only the corresponding vanilla SI-GNNs but also, in most cases, their counterparts trained without SC loss. This demonstrates the unique value of SC training: it makes the explainer more responsible in assigning importance scores, leading to more stable and higher-quality explanations.

Such a two-stage training scheme is not incidental but grounded in our motivation: explanation redundancy mainly arises because the explainer, when given sufficient budget, tends to behave irresponsibly. Therefore, SC loss only influences the explainer. The classifier is fine-tuned alongside the explainer because shifts in the explainer's behavior may alter the feature distribution of the graph representation, requiring the classifier to adapt correspondingly.

**Remark on Efficiency.** Compared to vanilla SI-GNNs, our two-stage framework introduces moderate additional training overhead (roughly doubling the training time) but yields substantial gains in explanation quality. Relative to EE—the first community-proposed solution for improving explanation quality—our method is significantly more efficient: since EE ensembles $K = 5$ independently trained models, our approach requires only $1/K$ (i.e., ~20%) of the memory or runtime cost at inference, while also avoiding the need to train multiple models.

Taken together, these results show that SC goes beyond faithfulness optimization: it provides a more efficient and effective mechanism than EE for mitigating redundancy, while also complementing EE—together they achieve better performance than either alone—and ultimately offers a scalable pathway to more trustworthy GNN explanations.

Table 5: Experimental results on SMGNN and GSAT (with different $\eta$). Notation (bold, color) follows the same convention as Table 1.

| Method | BA-2MOTIFS | | 3MR | | BENZENE | | MUTAGENICITY | |
|---|---|---|---|---|---|---|---|---|
| | ↓ SHD (%) | ↑ AUC (%) | ↓ SHD (%) | ↑ AUC (%) | ↓ SHD (%) | ↑ AUC (%) | ↓ SHD (%) | ↑ AUC (%) |
| SMGNN | 10.44±4.41 | 99.32±0.36 | 9.44±2.24 | 97.00±0.77 | 16.06±6.57 | 84.38±2.71 | 12.65±3.18 | 98.12±0.39 |
| SMGNN+EE | 4.99±1.48 | 99.59±0.16 | 5.20±0.72 | 98.25±0.21 | 8.55±2.79 | 91.38±0.96 | 6.21±1.20 | 98.96±0.16 |
| SMGNN+SC ($\eta$=0) | 8.46±3.48 | 99.26±0.49 | 13.01±3.02 | 98.72±0.35 | 8.30±2.39 | 88.72±1.25 | 9.68±2.34 | 97.04±0.74 |
| SMGNN+SC+EE ($\eta$=0) | 4.02±1.21 | 99.01±0.27 | 7.34±1.09 | 99.18±0.01 | 4.71±0.88 | 91.75±0.50 | 5.28±1.20 | 98.59±0.16 |
| SMGNN+SC ($\eta$=1) | 3.48±2.20 | 99.87±0.05 | 5.47±3.39 | 98.39±0.88 | 7.19±2.62 | 90.07±1.56 | 2.51±1.02 | 98.30±0.54 |
| SMGNN+SC+EE ($\eta$=1) | 1.52±1.03 | 99.90±0.02 | 3.05±1.22 | 99.25±0.02 | 4.14±0.82 | 92.20±0.43 | 1.44±0.38 | 98.96±0.17 |
| GSAT | 4.58±1.81 | 98.44±0.61 | 12.35±3.55 | 98.38±0.31 | 6.93±2.75 | 90.66±0.89 | 10.08±4.58 | 99.01±0.31 |
| GSAT+EE | 2.10±0.67 | 98.80±0.16 | 5.79±1.07 | 99.16±0.03 | 3.25±1.09 | 92.18±0.59 | 4.70±1.60 | 99.36±0.05 |
| GSAT+SC ($\eta$=0) | 5.03±1.78 | 85.22±4.26 | 9.52±2.82 | 99.17±0.23 | 5.30±0.90 | 92.79±0.47 | 6.71±0.65 | 99.04±0.33 |
| GSAT+SC+EE ($\eta$=0) | 2.25±0.88 | 86.33±3.54 | 4.48±1.17 | 99.42±0.01 | 2.49±0.31 | 93.60±0.13 | 3.03±0.29 | 99.29±0.08 |
| GSAT+SC ($\eta$=1) | 2.73±1.02 | 99.30±0.12 | 6.67±3.73 | 98.91±0.29 | 2.32±0.62 | 92.80±0.36 | 2.38±0.84 | 99.38±0.05 |
| GSAT+SC+EE ($\eta$=1) | 1.19±0.52 | 99.35±0.03 | 4.00±0.38 | 99.39±0.04 | 1.14±0.26 | 93.53±0.11 | 1.06±0.42 | 99.41±0.01 |
| | ↓ FID (%) | ↑ ACC (%) | ↓ FID (%) | ↑ ACC (%) | ↓ FID (%) | ↑ ACC (%) | ↓ FID (%) | ↑ ACC (%) |
| SMGNN | 0.50±0.92 | 95.50±12.52 | 1.90±1.01 | 97.30±1.31 | 2.07±0.62 | 91.12±0.62 | 1.72±1.09 | 89.53±0.99 |
| SMGNN+EE | – | 100.00±0.00 | – | 98.25±0.21 | – | 92.16±0.24 | – | 90.36±0.70 |
| SMGNN+SC ($\eta$=0) | 0.10±0.30 | 99.90±0.30 | 1.14±1.71 | 99.65±0.00 | 1.82±0.43 | 92.33±0.45 | 1.45±0.48 | 92.43±1.03 |
| SMGNN+SC+EE ($\eta$=0) | – | 100.00±0.00 | – | 99.65±0.00 | – | 92.81±0.18 | – | 93.39±0.47 |
| SMGNN+SC ($\eta$=1) | 0.00±0.00 | 99.80±0.60 | 0.00±0.00 | 99.65±0.00 | 0.97±0.59 | 92.57±0.44 | 0.61±0.54 | 91.18±1.07 |
| SMGNN+SC+EE ($\eta$=1) | – | 100.00±0.00 | – | 99.65±0.00 | – | 93.24±0.19 | – | 92.11±0.45 |
| GSAT | 0.00±0.00 | 100.00±0.00 | 0.90±0.28 | 98.55±0.80 | 1.80±0.84 | 91.48±0.87 | 1.11±0.61 | 92.43±1.00 |
| GSAT+EE | – | 100.00±0.00 | – | 99.15±0.24 | – | 92.17±0.28 | – | 93.00±0.44 |
| GSAT+SC ($\eta$=0) | 0.80±1.78 | 100.00±0.00 | 0.55±0.23 | 99.07±0.41 | 1.38±0.26 | 92.88±0.23 | 0.98±0.57 | 93.14±0.79 |
| GSAT+SC+EE ($\eta$=0) | – | 100.00±0.00 | – | 99.65±0.00 | – | 93.19±0.15 | – | 93.88±0.31 |
| GSAT+SC ($\eta$=1) | 0.00±0.00 | 100.00±0.00 | 0.31±0.29 | 99.55±0.16 | 0.74±0.14 | 92.31±0.32 | 0.17±0.23 | 93.48±0.48 |
| GSAT+SC+EE ($\eta$=1) | – | 100.00±0.00 | – | 99.65±0.00 | – | 92.61±0.18 | – | 93.76±0.37 |

Table 6: Experimental results on SMGNN and GSAT (with different training strategies). Notation (bold, color) follows the same convention as Table 1.

| Method | BA-2MOTIFS | | 3MR | | BENZENE | | MUTAGENICITY | |
|---|---|---|---|---|---|---|---|---|
| | ↓ SHD (%) | ↑ AUC (%) | ↓ SHD (%) | ↑ AUC (%) | ↓ SHD (%) | ↑ AUC (%) | ↓ SHD (%) | ↑ AUC (%) |
| SMGNN | 10.44±4.41 | 99.32±0.36 | 9.44±2.24 | 97.00±0.77 | 16.06±6.57 | 84.38±2.71 | 12.65±3.18 | 98.12±0.39 |
| SMGNN+SC (S1) | 7.71±6.14 | 99.31±0.45 | 15.55±6.33 | 95.38±1.51 | 19.33±5.43 | 73.59±7.69 | 8.58±4.01 | 97.30±0.67 |
| SMGNN+SC (S2) | 3.68±1.49 | 99.64±0.23 | 9.76±3.49 | 96.59±0.78 | 14.18±4.91 | 80.78±3.05 | 9.12±2.86 | 98.44±0.47 |
| SMGNN+SC (S3) | 8.28±10.61 | 99.61±0.89 | 12.11±3.13 | 97.60±0.48 | 10.91±2.66 | 81.83±2.54 | 7.69±3.96 | 97.91±0.42 |
| SMGNN+SC (S4) | 3.10±2.07 | 99.89±0.03 | 5.71±1.41 | 98.76±0.30 | 7.01±1.82 | 88.37±1.99 | 2.79±0.97 | 98.69±0.46 |
| SMGNN+SC | 3.48±2.20 | 99.87±0.05 | 5.47±3.39 | 98.39±0.88 | 7.19±2.62 | 90.07±1.56 | 2.51±1.02 | 98.30±0.54 |
| GSAT | 4.58±1.81 | 98.44±0.61 | 12.35±3.55 | 98.38±0.31 | 6.93±2.75 | 90.66±0.89 | 10.08±4.58 | 99.01±0.31 |
| GSAT+SC (S1) | 3.73±1.51 | 98.96±0.38 | 7.94±3.63 | 97.81±0.54 | 5.21±3.59 | 90.34±1.02 | 4.20±2.01 | 98.95±0.57 |
| GSAT+SC (S2) | 2.32±0.84 | 99.13±0.33 | 7.60±3.65 | 97.53±0.60 | 4.31±1.78 | 90.97±1.03 | 4.77±2.78 | 99.29±0.11 |
| GSAT+SC (S3) | 2.53±1.03 | 99.08±0.19 | 6.01±3.91 | 98.69±0.23 | 3.19±0.74 | 90.78±0.73 | 3.88±2.48 | 99.16±0.18 |
| GSAT+SC (S4) | 1.84±0.77 | 99.32±0.26 | 6.21±3.46 | 98.95±0.27 | 2.65±0.58 | 92.65±0.36 | 4.39±4.32 | 99.20±0.08 |
| GSAT+SC | 2.73±1.02 | 99.30±0.12 | 6.67±3.73 | 98.91±0.29 | 2.32±0.62 | 92.80±0.36 | 2.38±0.84 | 99.38±0.05 |
| | ↓ FID (%) | ↑ ACC (%) | ↓ FID (%) | ↑ ACC (%) | ↓ FID (%) | ↑ ACC (%) | ↓ FID (%) | ↑ ACC (%) |
| SMGNN | 0.50±0.92 | 95.50±12.52 | 1.90±1.01 | 97.30±1.31 | 2.07±0.62 | 91.12±0.62 | 1.72±1.09 | 89.53±0.99 |
| SMGNN+SC (S1) | 0.30±0.46 | 99.50±0.50 | 1.38±0.74 | 97.85±0.94 | 1.61±0.53 | 90.32±0.93 | 0.81±0.38 | 90.54±1.56 |
| SMGNN+SC (S2) | 0.20±0.40 | 99.90±0.30 | 1.28±0.31 | 98.24±0.57 | 1.67±0.59 | 90.66±1.37 | 1.15±0.66 | 89.49±1.16 |
| SMGNN+SC (S3) | 0.00±0.00 | 100.00±0.00 | 1.00±0.36 | 98.44±0.79 | 1.13±0.49 | 92.21±0.23 | 0.61±0.77 | 90.27±0.84 |
| SMGNN+SC (S4) | 0.00±0.00 | 100.00±0.00 | 0.45±0.49 | 99.27±0.42 | 1.66±0.47 | 92.45±0.36 | 0.68±0.64 | 90.78±1.50 |
| SMGNN+SC | 0.00±0.00 | 99.80±0.60 | 0.00±0.00 | 99.65±0.00 | 0.97±0.59 | 92.57±0.44 | 0.61±0.54 | 91.18±1.07 |
| GSAT | 0.00±0.00 | 100.00±0.00 | 0.90±0.28 | 98.55±0.80 | 1.80±0.84 | 91.48±0.87 | 1.11±0.61 | 92.43±1.00 |
| GSAT+SC (S1) | 0.50±0.50 | 99.60±0.49 | 0.76±0.40 | 98.34±0.46 | 1.19±0.47 | 92.03±0.53 | 0.17±0.23 | 92.53±1.17 |
| GSAT+SC (S2) | 0.40±0.49 | 99.60±0.49 | 1.00±0.57 | 98.10±0.52 | 1.08±0.17 | 92.36±0.37 | 0.24±0.30 | 92.87±0.87 |
| GSAT+SC (S3) | 0.10±0.00 | 99.90±0.30 | 0.03±0.10 | 99.62±0.10 | 1.03±0.35 | 92.83±0.54 | 0.17±0.23 | 93.11±0.88 |
| GSAT+SC (S4) | 0.00±0.00 | 100.00±0.00 | 0.10±0.16 | 99.62±0.10 | 1.05±0.19 | 92.45±0.36 | 0.20±0.22 | 93.24±0.83 |
| GSAT+SC | 0.00±0.00 | 100.00±0.00 | 0.31±0.29 | 99.55±0.16 | 0.74±0.14 | 92.31±0.32 | 0.17±0.23 | 93.48±0.48 |

### D.2.3 ABLATION: SHOULD THE ENCODER BE FROZEN DURING SC FINE-TUNING?

Our SC training framework freezes the GNN encoder during the fine-tuning stage (Step 2), so that the SC loss affects only the explainer (and the classifier that depends on its outputs). To examine whether this design choice is necessary, we evaluate four alternative configurations. These variants differ in (i) whether the encoder is fully or partially trainable during Step 2, and (ii) whether gradients from the SC loss are allowed to update the encoder. For clarity, we denote these settings as **S1–S4**:

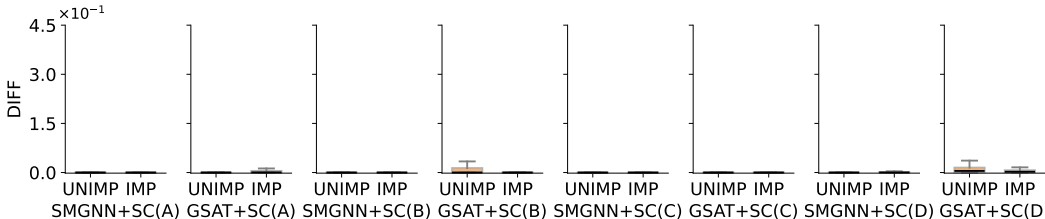

Figure 20: Self-inconsistency measured as the L1 difference (DIFF) between two explanations after SC training. Plotting convention follows Figure 1.

- **S1: Joint training, SC updates the encoder.** The encoder, explainer, and classifier are all trainable, and the SC loss backpropagates through the entire encoder.
- **S2: Joint training, SC does not update the encoder.** The encoder is trainable under the standard SI-GNN objective, but gradients from the SC loss are blocked before reaching the encoder.
- **S3: Last encoder layer trainable, SC updates the encoder.** Only the last encoder layer is unfrozen, and the SC loss is allowed to update this layer.
- **S4: Last encoder layer trainable, SC does not update the encoder.** The last encoder layer participates in standard SI-GNN optimization, while the SC loss remains restricted to the explainer.

Results in Table 6 provide the following practical guidance: (1) Do not allow the SC loss to update the encoder. (2) Do not fully unfreeze the encoder during SC fine-tuning. (3) We suggest freezing the encoder, which yields the most stable performance in practice.

## E PRACTICAL CONSIDERATIONS OF THE SC COEFFICIENT

This section provides a deeper discussion of the coefficient $\eta$ introduced by the SC loss. We address two aspects. First, in Section E.1, we show that incorporating SC does not alter the optimal explanatory subset, but instead reshapes the optimization landscape by discouraging redundant solutions. Second, in Section E.2, we analyze how to select $\eta$ in practice: by deriving upper bounds on the expected SC loss, we can restrict its search space during training, and we further report empirical findings that a simple choice $\eta = 1$ already works well across all datasets.

### E.1 PRESERVATION OF OPTIMAL EXPLANATIONS

Recall the original SI-GNN objective (Tai et al., 2025):

$$\max_{G_s \subseteq G,\ |G_s| \leq M} I(G_s; Y), \tag{26}$$

where $M$ is the budget and $G_s^*$ denotes the ground-truth explanatory subset.

With SC, the augmented training objective becomes:

$$\max_{G_s^{(1)} \subseteq G,\ G_s^{(2)} \subseteq G,\ |G_s^{(1)}| \leq M,\ |G_s^{(1)} - G_s^{(2)}| \leq \varepsilon} I(G_s^{(1)}; Y), \tag{27}$$

where $G_s^{(2)}$ is the explanation induced by feeding $G_s^{(1)}$ back into the model.

Since Figure 1 demonstrates that only unimportant edges exhibit self-inconsistency while important edges are inherently stable, it follows that the ground-truth explanation $G_s^*$ must also be self-consistent. Therefore, $G_s^*$ remains a feasible solution under the SC-augmented objective and achieves the same maximum MI as in the original one. In other words, SC does not alter the optimal explanation but reshapes the optimization landscape by discouraging redundant subsets that exploit unimportant edges. For comparison, we present the results after SC training in Figure 20.

## E.2 PRACTICAL GUIDELINES

The SC coefficient $\eta$ is the only additional hyperparameter introduced by our framework. During training, its search range can be guided by analyzing the expected value of the SC loss. Recall that the augmented objective (for SMGNN and GSAT) is:

$$\mathcal{L}_{\text{FT}} = \mathcal{L}_{\text{CE}} + \beta \cdot R(G_s) + \eta \cdot \mathcal{L}_{\text{SC}}. \tag{28}$$

The following result provides a theoretical upper bound on the expected self-consistency error.

**Proposition 1.** *Let $\theta$, $\phi$, and $\psi$ denote the parameters of the GNN encoder (fixed), explainer, and classifier, respectively. Define $\hat{\mathbb{E}}_{\mathcal{D}}$ as the empirical expectation over a set of samples $\mathcal{D}$. Let $\phi^* = \arg\min_\phi \hat{\mathbb{E}}_{\mathcal{D}}\left[\mathcal{L}_{\text{FT}}(G; \theta, \phi, \psi)\right]$, and $\tilde{\phi}$ denotes any perfectly self-consistent solution that satisfy $\tilde{\phi} = \arg\min_\phi \hat{\mathbb{E}}_{\mathcal{D}}\left[\mathcal{L}_{\text{SC}}(G_s^{(1)}, G_s^{(2)}; \theta, \phi)\right]$. Then, we have:*

$$\hat{\mathbb{E}}_{\mathcal{D}}\left[\mathcal{L}_{\text{SC}}(G_s^{(1)}, G_s^{(2)}; \theta, \phi^*)\right] \leq \frac{\log C + \Omega}{\eta}, \tag{29}$$

*where $C$ is the number of label types, and $\Omega$ is the maximum value of $\beta \cdot R(G_s)$.*

*Proof.* Since $\phi^*$ is the minimizer of $\hat{\mathbb{E}}_{\mathcal{D}}\left[\mathcal{L}_{\text{FT}}(G; \theta, \phi, \psi)\right]$, we have:

$$\hat{\mathbb{E}}_{\mathcal{D}}\left[\mathcal{L}_{\text{GE}}(G; \theta, \phi^*, \psi) + \eta \cdot \mathcal{L}_{\text{SC}}(G_s^{(1)}, G_s^{(2)}; \theta, \phi^*)\right]$$
$$\leq \hat{\mathbb{E}}_{\mathcal{D}}\left[\mathcal{L}_{\text{GE}}(G; \theta, \tilde{\phi}, \psi) + \eta \cdot \mathcal{L}_{\text{SC}}(G_s^{(1)}, G_s^{(2)}; \theta, \tilde{\phi})\right]. \tag{30}$$

Since $\mathcal{L}_{\text{SC}}(G_s^{(1)}, G_s^{(2)}; \theta, \tilde{\phi}) = 0$, we have:

$$\hat{\mathbb{E}}_{\mathcal{D}}\left[\mathcal{L}_{\text{GE}}(G; \theta, \phi^*, \psi) + \eta \cdot \mathcal{L}_{\text{SC}}(G_s^{(1)}, G_s^{(2)}; \theta, \phi^*)\right] \leq \hat{\mathbb{E}}_{\mathcal{D}}\left[\mathcal{L}_{\text{GE}}(G; \theta, \tilde{\phi}, \psi)\right]. \tag{31}$$

Due to the non-negative property of $\mathcal{L}_{\text{GE}}$, the above inequality can be further simplified:

$$\hat{\mathbb{E}}_{\mathcal{D}}\left[\mathcal{L}_{\text{GE}}(G; \theta, \phi^*, \psi) + \eta \cdot \mathcal{L}_{\text{SC}}(G_s^{(1)}, G_s^{(2)}; \theta, \phi^*)\right] \leq \hat{\mathbb{E}}_{\mathcal{D}}\left[\mathcal{L}_{\text{GE}}(G; \theta, \tilde{\phi}, \psi)\right] \tag{32}$$

$$\hat{\mathbb{E}}_{\mathcal{D}}\left[\eta \cdot \mathcal{L}_{\text{SC}}(G_s^{(1)}, G_s^{(2)}; \theta, \phi^*)\right] \leq \hat{\mathbb{E}}_{\mathcal{D}}\left[\mathcal{L}_{\text{GE}}(G; \theta, \tilde{\phi}, \psi) - \mathcal{L}_{\text{GE}}(G; \theta, \phi^*, \psi)\right] \tag{33}$$

$$\hat{\mathbb{E}}_{\mathcal{D}}\left[\eta \cdot \mathcal{L}_{\text{SC}}(G_s^{(1)}, G_s^{(2)}; \theta, \phi^*)\right] \leq \hat{\mathbb{E}}_{\mathcal{D}}\left[\mathcal{L}_{\text{GE}}(G; \theta, \tilde{\phi}, \psi)\right] \tag{34}$$

$$\hat{\mathbb{E}}_{\mathcal{D}}\left[\mathcal{L}_{\text{SC}}(G_s^{(1)}, G_s^{(2)}; \theta, \phi^*)\right] \leq \hat{\mathbb{E}}_{\mathcal{D}}\left[\frac{\mathcal{L}_{\text{GE}}(G; \theta, \tilde{\phi}, \psi)}{\eta}\right]. \tag{35}$$

Suppose the number of samples in different classes is the same. The worst-case for the GNN explainer happens when it discards all edges relevant to the label, but preserves all irrelevant edges. Under such circumstances, we have a perfectly self-consistent explainer but the downstream classifier cannot obtain any information from it. Thus, we have:

$$\mathcal{L}_{\text{GE}} = \mathcal{L}_{\text{CE}} + \beta \cdot R(G_s) \leq \log C + \beta \cdot R(G_s). \tag{36}$$

Substituting Equation (36) into Equation (35), we can get:

$$\hat{\mathbb{E}}_{\mathcal{D}}\left[\mathcal{L}_{\text{SC}}(G_s^{(1)}, G_s^{(2)}; \theta, \phi^*)\right] \leq \frac{\log C + \Omega}{\eta}, \tag{37}$$

where $\Omega$ is the maximum value of $\beta \cdot R(G_s)$. $\qquad\square$

For SMGNN that uses the sparsity loss, the worst case is:

$$\max_{G_s} \beta \cdot R(G_s) = \beta \cdot \frac{|G_s|}{|G|} \leq \beta. \tag{38}$$

Thus, for SMGNN, we have:

$$\hat{\mathbb{E}}_{\mathcal{D}}\left[\mathcal{L}_{\text{SC}}(G_s^{(1)}, G_s^{(2)}; \theta, \phi^*)\right] \leq \frac{\log C + \beta}{\eta}. \tag{39}$$

For GSAT that uses KL divergence to constrain $G_s$, the worst case is:

$$\max_{G_s} \beta \cdot R(G_s) = \frac{\beta}{|G|} \sum_{\alpha_{ij} \in \mathcal{E}} \alpha_{ij} \log \frac{\alpha_{ij}}{r} + (1 - \alpha_{ij}) \log \frac{1 - \alpha_{ij}}{1 - r}$$

$$\leq \beta \cdot \max_{\alpha_{ij}} \left[\alpha_{ij} \log \frac{\alpha_{ij}}{r} + (1 - \alpha_{ij}) \log \frac{1 - \alpha_{ij}}{1 - r}\right], \tag{40}$$

where $r \in [0, 1]$ is a hyperparameter controlling the conciseness of the GNN explanation. To calculate the maximum value of Equation (40), we need to calculate the first and second derivatives of $p(\cdot)$. Setting the derivative equal to zero, we solve $\frac{\alpha(1-r)}{r(1-\alpha)} = 1$, which yields $\alpha = r$. The second derivative $\frac{1}{\alpha(1-\alpha)}$ is always positive, indicating that the function attains a local minimum at $\alpha = r$ and maximum at end points. So the maximum value is: $\beta \cdot \max(\log(\frac{1}{r}), \log(\frac{1}{1-r}))$. Thus, for GSAT, we have:

$$\hat{\mathbb{E}}_{\mathcal{D}}\left[\mathcal{L}_{\text{SC}}(G_s^{(1)}, G_s^{(2)}; \theta, \phi^*)\right] \leq \frac{\log C + \beta \cdot \max(\log(\frac{1}{r}), \log(\frac{1}{1-r}))}{\eta}. \tag{41}$$

In practice, $r$ will not be set too large or too small—too small $r$ means that most edges will be discarded, which makes the training process unstable, and $r$ that is too large cannot provide valuable explanations. In our experiments, we set $r = 0.5$, so we have: $\eta \leq \frac{\log C + \beta \cdot \log 2}{\delta}$.

Given the tolerance factor $\delta$, we first formalize a candidate set $\mathcal{S}(\eta), \eta \in (0, \frac{\log C + \Omega}{\delta}]$. Then we select $\eta$ from $\mathcal{S}(\eta)$ based on validation performance. In our experiments, we find that $\eta = 1$ already yields good enough results on all datasets.

## F    THE USE OF LARGE LANGUAGE MODELS

ChatGPT 5 were used as an assistive tool during the preparation of this paper. Their role was limited to improving the clarity and readability of the text (e.g., language polishing) and checking the presentation of mathematical derivations. All research ideas, methodological designs, experiments, analyses, and conclusions were solely developed and validated by the authors, who take full responsibility for the content of this work.

