# OpenReview forum: "Self-Consistency Improves the Trustworthiness of Self-Interpretable GNNs"
_ICLR.cc/2026/Conference — ICLR 2026 Poster_

### Official Review · Reviewer_7o1e · 2025-10-17

**Soundness:** 4
**Presentation:** 4
**Contribution:** 4
**Rating:** 8
**Confidence:** 3

**Summary:**

This paper introduce a simple, model-agnostic self-consistency (SC) post-training strategy for pretrained Self-Interpretable GNNs (SI-GNNs).
Rigorous analysis demonstrates that faithfulness can be explicitly optimized through the strategy, and such optimization genuinely improves explanation quality.
Experiments show that SC consistently improves explanation quality across multiple dimensions and benchmarks, offering an effective and scalable pathway to more trustworthy GNN explanations.

**Strengths:**

- The paper provides a fresh perspective (Self-Consistency) to improves the trustworthiness of self-interpretable GNNs, , the theoretical and empirical verification of the key assumptions is sufficient, there are no obvious flaws.
- The method demonstrates strong explanation performance across multiple tasks, improving various types of self-Interpretable GNNs.
- The paper is well-organized and easy to follow.

**Weaknesses:**

1. During the fine-tuning phase of SC (Section 3.1), this work freezes the GNN encoder and assumes that "the encoder representation is already optimal, and only the explainer needs optimization". However, this work fails to validate this design and could further compare it with a control group where "the encoder is not frozen".
2. Has the author conducted statistical significance tests? What percentage of the experimental results significantly outperform each baseline? This is crucial for understanding the overall performance of SC.

**Questions:**

Please refer to the weaknesses for suggestions. This article is highly accomplished, and the weaknesses mentioned above will not change my positive evaluation.

---

> ### Author Response · Authors · 2025-11-14
>
> Thank you very much for the thoughtful and constructive reviews. Below, we provide point-by-point responses to your questions.
>
> > During the fine-tuning phase of SC (Section 3.1)...
>
> Thank you for raising this important point. In the revised version, we have added a new subsection D.2.3 (Ablation: Should the Encoder Be Frozen During SC Fine-Tuning?) in the Appendix.
>
> To evaluate this design choice, we conduct four controlled variants:
> - S1: Joint training, SC updates the encoder
> - S2: Joint training, SC does not update the encoder
> - S3: Last encoder layer trainable, SC updates the encoder (asked by Reviewer hKUr)
> - S4: Last encoder layer trainable, SC does not update the encoder (asked by Reviewer hKUr)
>
> (These variants correspond to the configurations reported as SI-GNN+SC (S1-S4) in Table 6.)
>
> Results provide the following practical guidance: (1) Do not allow the SC loss to update the encoder. (2) Do not fully unfreeze the encoder during SC fine-tuning. (3) We suggest freezing the encoder, which yields the most stable performance in practice.
>
> > Has the author conducted statistical significance tests...
>
> Thank you for the suggestion. Following your comment, we performed Mann–Whitney U test. Across four SI-GNNs, four datasets, and four metrics (64 cases in total), SC improves explanation quality in 92.2% (59) cases.
>
> Among these 59 positive cases:
> - 89.8% (53) show statistically significant improvement at p < 0.05,
> - 80.0% (46) show statistically significant improvement at p < 0.01.
> - 72.9% (43) show statistically significant improvement at p < 0.005.
> - 57.6% (34) show statistically significant improvement at p < 0.001.
>
> These results confirm that the performance gains brought by SC are not only consistent but also statistically significant. We have included all 64 p-values in the supplementary material (see test.csv if interested).
>
> Thank you once again for the thoughtful comments and for your strong endorsement of our work; it means a lot to us :)

---

> > ### Comment · Reviewer_7o1e · 2025-11-19
> >
> > What can I say, man. You have addressed all my concerns, so I'll have to consider changing my rating to 10/10 if I see that the other reviewers' concerns are also properly addressed.

---

> ### Author Response · Authors · 2025-11-19
>
> Thank you so much for the incredibly kind message, and we truly appreciate your support!

---

### Official Review · Reviewer_hKUr · 2025-10-28

**Soundness:** 3
**Presentation:** 3
**Contribution:** 3
**Rating:** 4
**Confidence:** 5

**Summary:**

The paper studies a common mismatch in self-interpretable GNNs (SI-GNNs): models are trained with classification plus conciseness regularization, but are evaluated by faithfulness (whether the highlighted subgraph alone reproduces the prediction). The authors argue that faithfulness is intrinsically linked to self-consistency (SC)—the explanation produced on a graph should be reproduced when the model is run on its own explanation. They propose a simple, model-agnostic training strategy that adds a self-consistency loss during a short fine-tuning stage where the GNN encoder is frozen and only the explainer and classifier are updated, yielding a final objective.
They analyze how enforcing self-consistency drives importance scores toward a few “near fixed levels,” and how this interacts with conciseness regularization (CR) such as sparsity (GISST) or MI constraints (GSAT) to suppress unimportant edges while preserving important ones.
On four benchmarks (BA-2MOTIFS, 3MR, BENZENE, MUTAGENICITY) and four SI-GNN families (GISST, GSAT, GAT, CAL), SC improves consistency, faithfulness, explanation accuracy, and informativeness; it also complements explanation ensembling (EE) and is substantially more efficient at inference. SC alone may hurt when CR is absent (GAT/CAL), but adding CR stabilizes and recovers gains. The method requires no architectural change and adds one low-sensitivity hyperparameter ($\eta$) .

**Strengths:**

1. Reframes faithfulness optimization as enforcing *self-consistency* during training and connects it to redundancy on unimportant edges; provides a model-agnostic, plug-in loss with a two-stage schedule that freezes the encoder  .
2. Extensive experiments on four datasets and four SI-GNN families. Clear reporting that SC alone can fail without CR, plus the complementary effect with EE. Solid ablations on ($\beta$) and ($\eta$) sensitivity and two-stage training to rule out confounds  .
3. The training procedure and metrics are well defined; the figures make the stability effect tangible, and the paper carefully explains when SC helps and when it needs CR support .
4. Practical because it needs no architecture changes, adds a single hyperparameter of low sensitivity, and improves multiple explanation dimensions while being more efficient than EE at inference time .

**Weaknesses:**

1. The two-stage setup assumes that Step 1 already produces explanations covering the ground-truth rationale. Step 2 then removes redundancy. If Step 1 misses key parts of the rationale, SC may stabilize an incomplete subset rather than recover it. The paper should explicitly state this assumption and test its robustness.

2. Adding more initial explanations (e.g., multiple seeds or ensembling in Step 1) could alleviate this limitation. The gains seen in Table 1 when combining with EE support this interpretation.

3. All datasets are small; results on large-scale or node-level tasks would clarify generality.

4. Limited theory: The fixed-level analysis is illustrative but not formal; it would be valuable to characterize conditions guaranteeing convergence toward faithful explanations.

5. Incomplete metric coverage: Improvements on FID− are strong, while FID+ results remain mixed; additional discussion would improve completeness.

**Questions:**

1. What happens if the Step 1 SI-GNN misses part of the ground-truth rationale? Please evaluate robustness by ablating a portion of important edges before Step 2 and reporting whether SC can recover faithfulness or simply stabilize incomplete masks.

2. How many models are produced overall, and which one generates explanations? My understanding is that only a single model—the Step 2 checkpoint—is used at inference. If multiple initial explanations are generated in Step 1, are they ensembled before Step 2 or during inference?

3. Consider generating multiple Step 1 explanations and comparing unions or averaged masks before Step 2. This could test whether broader initial coverage yields better results.

4. Since SC alone can hurt for GAT/CAL, what is the weakest form of conciseness regularization ((L_1) or entropy penalty) that stabilizes training?

5. Could encoder-side regularization, such as partial Mixup on masked graphs, reduce encoder distribution shift and improve FID+?

6. Would allowing the last encoder layer to update during Step 2 better adapt to explainer changes without destabilizing learned representations?

---

> ### Author Response · Authors · 2025-11-14
> **Part 1/2**
>
> Thank you very much for the detailed and thoughtful review. We sincerely appreciate the time you invested in evaluating our work. Before providing point-by-point responses to your questions, we would like to clarify three important points regarding how SC works:
> - According to Eq.(17)-(19), there is no guarantee that $G_s^{(2)} \subseteq G_s^{(1)}$.
> - $G_s^{(1)}$ is typically redundant, not incomplete. Following both Tai et al. and our experiments, existing SI-GNNs on the studied benchmark datasets consistently include all informative edges; the main issue is that some uninformative edges also receive high scores. Self-inconsistency occurs on part of these uninformative edges, and SC training is designed to process these unstable, uninformative edges.
> - SC+EE is "SC applied before EE, not after".
>   - +SC = train a single SI-GNN with standard loss plus SC loss.
>   - +SC+EE = first train multiple SC-trained models (different seeds), and then ensemble their explanations at inference.
>
> > The two-stage setup assumes that Step 1 already produces explanations covering the ground-truth rationale...What happens if the Step 1 SI-GNN misses part of the ground-truth rationale...
>
> Step 1 almost always covers the full ground-truth rationale on all four benchmarks. Cases where step 1 misses an informative edge are extremely rare. When such rare cases occur, we observe both possibilities:
> - SC may recover the missing informative edge if its step-2 score is high.
> - SC may not recover the missing informative edge if its step-2 score is consistently low.
>
> Whether a missed informative edge is recovered depends on whether it exhibits self-inconsistency across passes. We have added case visualizations (SVG files, BENZENE dataset, GISST vs. GISST+SC) to the supplemental material.
>
> > Adding more initial explanations (e.g., multiple seeds or ensembling in Step 1) could alleviate this limitation…
>
> SC never ensembles step-1 explanations; EE operates after SC.
>
> > How many models are produced overall, and which one generate explanations...
>
> SC is strictly a single-model procedure. For +SC, we train one SI-GNN and use the step-2 checkpoint for inference. For +SC+EE, we apply EE after SC (at inference time).
>
> > Consider generating multiple Step 1 explanations and comparing unions or averaged masks before Step 2...
>
> Thank you for the comment. In SC, both $G_s^{(1)}$ and $G_s^{(2)}$ must be produced by the same model, since we fine-tune the explainer to be self-consistent with its own explanation.
>
> > All datasets are small...
>
> Thank you for raising this excellent point. As researchers, we share the same concern and are also constrained by the current limitations of the SI-GNN evaluation ecosystem. Since we follow Tai et al. and use AUC to assess explanation quality, the range of feasible datasets becomes very restricted: (1) most graph benchmarks do not provide annotated ground-truth explanations, and (2) even when annotations exist, we must ensure they are reliable, which requires additional verification procedures [1]. Given that current removal-based evaluation still suffers from well-known issues (please see our response to Reviewer AnRn), we view the development of scalable and reliable explanation datasets as an important and highly promising direction for future work (and we are actively exploring more reliable evaluation metrics and establishing more comprehensive evaluation frameworks).
>
> > Limited theory: The fixed-level analysis is illustrative but not formal; it would be valuable to...
>
> Thank you for the suggestion. To be honest, providing a formal convergence guarantee would require strong assumptions that we cannot rigorously justify at this stage; we leave such theoretical characterization as a direction for future work.
>
> > Incomplete metric coverage...
>
> Thank you for the comment. As discussed in Appendix D.2 (updated as D.2.1 in the revision), we argue that FID+ should not be over-interpreted in SI-GNNs due to its inherent OOD nature — models never see $G \setminus G_s$ during training, so FID+ reflects a mixture of explanation quality and the model's generalizability [2,3]. We provide a detailed discussion of this issue in our response to Reviewer AnRn; please refer to it if interested.
>
> > Since SC alone can hurt for GAT/CAL, what is the weakest form...
>
> In our implementation, we simply adopt the same sparsity loss used in GISST, with the same small coefficient $0.05$ (following GISST's setting).
>
> **(To be continued...)**

---

> ### Author Response · Authors · 2025-11-14
> **Part 2/2**
>
> > Could encoder-side regularization, such as partial Mixup on masked graphs, reduce encoder distribution shift and improve FID+?
>
> Thank you for the insightful comment. In principle, encoder–side regularization such as partial Mixup could reduce distribution shift, and similar ideas have indeed been explored in post–hoc explainers (e.g., MixupExplainer [4]; ProxyExplainer [5]). We do not adopt such techniques in our experiments for three reasons:
> - **Defining a "clean" Mixup target in practice is non-trivial.** Mixup operation may unintentionally introduce label information, which confounds FID+ evaluation.
> - **It is unclear whether Mixup fundamentally eliminates the OOD issue.** As long as distribution shifts remain, FID+ continues to be an unreliable metric.
> - **We already rely on a comprehensive and well-established evaluation protocol (accuracy, consistency, faithfulness, informativeness) [2].** Given the known limitations of FID+, using complex OOD-mitigation techniques (such as training a VGAE in ProxyExplainer) adds substantial overhead while providing limited benefit.
>
> PS: Some recent works propose alternative metrics that aim to improve the reliability of faithfulness evaluation (e.g., RFID [6]). RFID can be viewed as a trade-off between reducing OOD effects and preserving evaluation validity.
>
> > Would allowing the last encoder layer to update during Step 2 better adapt to explainer changes without destabilizing learned representations?
>
> Thank you for raising this important point. In the revised version, we have added a new subsection D.2.3 (Ablation: Should the Encoder Be Frozen During SC Fine-Tuning?) in the Appendix.
>
> To evaluate this design choice, we conduct four controlled variants:
> - S1: Joint training, SC updates the encoder (asked by Reviewer 7o1e)
> - S2: Joint training, SC does not update the encoder (asked by Reviewer 7o1e)
> - S3: Last encoder layer trainable, SC updates the encoder
> - S4: Last encoder layer trainable, SC does not update the encoder
>
> (These variants correspond to the configurations reported as SI-GNN+SC (S1-S4) in Table 6.)
>
> Results provide the following practical guidance: (1) Do not allow the SC loss to update the encoder. (2) Do not fully unfreeze the encoder during SC fine-tuning. (3) We suggest freezing the encoder, which yields the most stable performance in practice.
>
> Regarding your specific question—whether allowing only the last encoder layer to update during step 2 would better adapt to explainer changes: unfreezing only the last encoder layer performs similarly to the frozen-encoder setup.
>
> We hope that our responses address your concerns, and we sincerely appreciate your time and effort in reviewing our work :)
>
> Ref:
> *[1] Faber et al., When comparing to ground truth is wrong: On evaluating gnn explanation methods, KDD, 2021*
> *[2] Azzolin et al. Reconsidering faithfulness in regular, self-explainable and domain invariant gnns, ICLR, 2025*
> *[3] Tai et al., Redundancy undermines the trustworthiness of self-interpretable gnns, ICML, 2025*
> *[4] Zhang et al., MixupExplainer: Generalizing explanations for graph neural networks with data augmentation, KDD, 2023*
> *[5] Chen et al., Interpreting graph neural networks with in-distributed proxies, ICML, 2024*
> *[6] Zheng et al., Towards robust fidelity for evaluating explainability of graph neural networks, ICLR, 2024*

---

### Official Review · Reviewer_AnRn · 2025-11-01

**Soundness:** 3
**Presentation:** 3
**Contribution:** 2
**Rating:** 4
**Confidence:** 4

**Summary:**

The authors address a relevant and widespread problem in GNN (and SE-GNN in particular), which is the consistency and reliability of the explanations being extracted, and propose an approach to encourage self-consistency of explanations at training time.

**Strengths:**

Trying to directly enforce consistency at training time seems like a sensible direction.

Experimental results confirm improvements and provide insights into explanations quality and robustness.

**Weaknesses:**

The problem of consistency of explanations was already addressed in the work by Tai et al, where they propose a simple esamble strategy (EE). The novelty of the work is thus not dramatic. The authors show that their approach improves over EE (apart from being clearly much faster), and their combination further improves results. The rationale for these results is unclear to me. Is there any substantial difference in what the approaches tackle justifying this? Is this a matter of hyperparameter choice? This is important to correctly evaluate the relevance of the contribution.


There are plenty of notions of faithfulness of explanations, but a key aspect is that one should measure both sufficiency (e.g. with FID-) and precision (e.g. with FID+). I thus think the authors should include FID+ (or similar metrics) to get the full picture. Given that the proposed SC component is combined with conciseness regularization (CR), I do not expect this to undermine the utility of the approach, but it would allow to get a better picture of its contribution, especially when seeing the FID+ results without CR (table 3).

**Questions:**

Can you clearly motivate the performance difference between SC and EE? aren't they basically optimizing for the same objective?

Can you add FID+ results and comment them to better understand the interplay between SC and CR?

---

> ### Author Response · Authors · 2025-11-14
>
> Thank you very much for the thoughtful and constructive reviews. Below, we provide point-by-point responses to your questions.
>
> > The problem of consistency of explanations was already addressed in the work by Tai et al...Can you clearly motivate the performance difference between SC and EE...
>
> We would like to clarify that our work is conceptually and technically distinct from Tai et al.
>
> - **Different motivations.** Tai et al. study cross-model explanation inconsistency, asking **why multiple independently trained SI-GNNs produce different explanations**. Their analysis shows that redundancy leads to (cross-model) variance, and EE is designed based on this phenomenon. Our work is motivated by a completely different research question: **Can SI-GNNs be directly optimized for faithfulness, and is such optimization truly necessary?** During the investigation, we found that faithfulness optimization has a potential connection to the explanation redundancy observed by Tai et al: faithfulness optimization naturally connects to (single-model) self-consistency, and that existing SI-GNNs exhibit self-inconsistency, especially on uninformative, redundant edges. Therefore, the statement that "the problem of consistency was already addressed by Tai et al." does not reflect our contribution: we study single-model faithfulness-oriented optimization, not cross-model explanation inconsistency.
> - **Different mechanisms and complementary effects.** EE focuses on redundant edges that exhibit cross-model high variance. SC focuses on redundant edges that exhibit single-model self-inconsistency. Since these are not the same subset of edges, neither strategy can eliminate redundancy alone, and their complementary nature explains why SC + EE performs best.
> - **Practical advantages of SC.** EE requires multiple SI-GNNs, making it computationally (both training and inference) expensive and incompatible with single-model metrics such as Fidelity. SC operates entirely within one model, is far more efficient, and remains compatible with all evaluation metrics.
>
> > There are plenty of notions of faithfulness of explanations...Can you add FID+ results and comment them to better understand the interplay between SC and CR?
>
> Thank you for the suggestion. Our initial submission (Appendix D.2; now D.2.1) already included FID+ results for GISST and GSAT, where SC showed consistent gains on FID- but mixed effects on FID+. Following your suggestion, we have added GAT and CAL, and the observation remains unchanged. Below, we explain why FID+ should be interpreted cautiously in SI-GNN evaluation, and why these mixed results do not contradict the effectiveness of SC.
> - **FID+ suffers from the OOD issue in SI-GNNs.** To minimize confounding factors, consider BA-2MOTIFS dataset where all four SI-GNNs achieve high AUC and ACC. According to Tai et al. (and our experiments), explanations cover all informative edges, and the main issue is redundancy. Thus, ideally, FID− should be low (because $G_s$ contains all informative edges) and FID+ should be high (because removing $G_s$ leaves only uninformative edges). For the three SI-GNNs (GAT, GISST, GSAT) that do see $G_s$ during training: FID− is good (0.0%–2.5%), but FID+ is much worse (27.4%–46.4%). This is exactly what the OOD effect predicts: during training the model sees $G_s$ but never sees $G \setminus G_s$, so FID+ is evaluated entirely on OOD graphs. (PS: For CAL, its methodological design leads to OOD issue even for FID−, which explains its higher value. To avoid a tangential digression, we do not elaborate here, but we would be happy to discuss more details if you are interested.)
> - **Even ignoring the OOD issue, FID metrics still have some limitations:** (1) FID+ is insensitive to redundancy (and FID- may even benefit from it). For example, FID- achieves the best when $G_s = G$ (trivial explanation). Because SC implicitly targets redundancy, relying solely on FID would obscure the very phenomenon we aim to address. (2) The design of the SC loss implicitly encourages lower FID−, so evaluating SC with FID- would violate the principle of "not letting the method grade itself." This is why complementary metrics such as AUC, SHD, and ACC are needed for a comprehensive evaluation. (3) AUC is a more reliable precision metric than FID+ (we feel that FID+ is not a precision metric). As advocated by Faber et al. [1], when annotated explanations are justified, AUC is preferred, because removal-based evaluation (FID, etc.) conflates explanation quality with model generalizability.
>
> For these reasons, we follow Tai et al. and adopt AUC as the primary metric and use FID, SHD, and ACC as complementary metrics to comprehensively evaluate the effectiveness of SC.
>
> We hope that our responses address your concerns, and we sincerely appreciate your time and effort in reviewing our work :)
>
> Ref:
> *[1] Faber et al., When comparing to ground truth is wrong: On evaluating gnn explanation methods, KDD, 2021*

---

> > ### Comment · Reviewer_AnRn · 2025-11-25
> >
> > Thanks a lot for your detailed feedback and for including FID+ in the evaluation. I do agree that this metric is far from perfect, but FID- alone gives a very partial picture.
> >
> > About the relationship with Tai et al, it is clear to me that you started from a different research question, I am not questioning it. My point is rather tied to the close relationship between inconsistency across models and across data within a model. I think that a deeper understanding of the similarities and differences between these two aspects would substantially strengthen the paper.
> >
> > That said, I am willing to raise my score over the acceptance bar given the changes you made.

---

> ### Author Response · Authors · 2025-11-25
>
> Thank you very much for your constructive follow-up and for raising your score. We truly appreciate it!
>
> Both of the points you raised are insightful, and we would be happy to share some additional thoughts on them below.
>
> (1) FID metrics
>
> Regarding FID metrics, we fully agree that relying on FID− alone would provide an incomplete picture. In our work, FID− is not intended to serve as the primary evaluation metric — its role is mainly diagnostic, to verify that SC indeed optimizes faithfulness as designed. The evaluation of explanation quality is based on the four complementary dimensions: accuracy, consistency, faithfulness, and informativeness. To be honest, given the limitations of both AUC and FID, we are actively exploring more comprehensive evaluation framework (though progress is slow).
>
> (2) Relationship Between Cross-Model Inconsistency and Single-Model Self-Inconsistency
>
> We believe that these two forms of inconsistency share a common underlying cause: the sufficient explanation budget, which manifests as redundancy. Building on this, we propose a hypothesis that helps explain why both forms arise and why their identified subsets only partially overlap.
>
> When the explanation budget is sufficient, informative edges acquire reinforcing signals, enabling the explainer to consistently assign them high importance. In contrast, uninformative edges lack such reinforcing signals (and a sufficient budget does not introduce any suppressive signal that would push their scores down), so their importance estimates rely heavily on contextual factors.
>
> Under this assumption: (1) Cross-model inconsistency arises because different random initializations lead to different model parameterizations and, consequently, different contextual patterns for uninformative edges. (2) Single-model self-inconsistency arises because the neighborhood structure of an edge changes between the first and second passes ($G$ and $G_s^{(1)}$, respectively), which influences the learned edge representation, and thus its score.
>
> This perspective suggests that the two forms of inconsistency are triggered by different sources of variation — across parameter spaces in one case and across masked graph inputs in the other. Since this remains a hypothesis rather than a formal theory, we view developing a more rigorous understanding as an important direction for future work.
>
> If you have different thoughts, we would be very happy to continue the discussion, and we sincerely appreciate your insightful questions and feedback. Thanks!

---

### Official Review · Reviewer_25Yx · 2025-11-02

**Soundness:** 3
**Presentation:** 4
**Contribution:** 3
**Rating:** 8
**Confidence:** 3

**Summary:**

The authors propose a simple but sensible model-agnostic strategy for improving the faithfulness of self-explainable GNNs. The idea is to penalize the model for "changing its mind" whenever fed with a local explanation of a certain prediction during training. The technique is validated using four approaches and compared
against an alternative (explanation ensembling) on four datasets, with encouraging
results.

**Strengths:**

- **Originality**: The idea behind the penalty is aligned with existing results,
  but otherwise original -- to the best of my knowledge.

- **Quality**: The proposed idea is sensible. The empirical setup is also good
  -- the choice of datasets, metrics and competitors all look good. I appreciate
  how the authors clearly distinguish between faithfulness (measured with Fid-)
  and plausibility/explanation accuracy. This is surprisingly rare in the
  literature.

- **Clarity**: The text is very clear and well structured. The visualizations
  are helpful.

- **Significance**: The contribution is welcome and I think it bridges a
  serious gap in the literature, by making theoretical insights practical.
  The fact that the approach is model agnostic also helps.

TL;DR: good paper, I like the idea and the execution.

**Weaknesses:**

- **Originality**: As I mentioned, I believe the proposed technique follows
  naturally from existing results (eg Azzolin et al, who the authors mention).
  This is however not a major issue for me - the penalty, as I mentioned,
  is novel.

- **Clarity**: My only real complaint is that Table 1 has too many colors, making
  it diffult to focus on what's really relevant. I'd suggest to tone it down.

**Questions:**

None.

---

> ### Author Response · Authors · 2025-11-14
>
> Thank you very much for the thoughtful and encouraging review. Following your suggestion, we have revised Table 1 to highlight only the best and second-best results (we would love to consider any suggestions you may have). We sincerely appreciate your recognition and constructive feedback :)

---

> > ### Comment · Reviewer_25Yx · 2025-11-14
> > **Suggestion**
> >
> > I have one minor suggestion that I forgot to include in my review: Please make sure to highlight in the text that you measure fidelity/faithfulness using Fid-. Not all faithfulness metrics behave the same, so it's a good idea to clarify the one your are using explicitly.

---

> ### Author Response · Authors · 2025-11-14
>
> Thank you for the helpful suggestion. We fully agree that different faithfulness metrics behave differently and should be stated explicitly. As noted on page 6, line 318, we use FID− to evaluate faithfulness, and Appendix C provides the formal definitions of both FID− and FID+, along with an explanation of why FID+ is not used in the main paper (while we still report FID+ results Appendix D.2 for completeness).

---

> > ### Comment · Reviewer_25Yx · 2025-11-17
> > **Reply**
> >
> > Whoops, I was under the impression FID- was only hinted at in the appendix. My bad. I retract my suggestion.

---

> > > ### Author Response · Authors · 2025-11-17
> > >
> > > Thank you for the follow-up! No worries at all, and we sincerely appreciate your recognition of our paper. If you have any further suggestions at any time, please feel free to share them. We would be very happy to hear your thoughts.

---

### Author Response · Authors · 2025-11-19
**PDF Update Note (Nov 19) — Minor Correction to FID Results for CAL**

**PDF Update Note (Nov 19) — Minor Correction to FID Results for CAL**

Dear Reviewers,

Following Reviewer AnRn's suggestion, we updated the PDF on Nov 14 to include FID+ results for GAT and CAL. During a careful check today, we identified a small implementation bug related to how the complement of the explanation was constructed for CAL during the FID evaluation. We have corrected this and updated the following CAL (series) results:

- Table 3: FID−
- Table 4: FID− and FID+

**Impact on the paper**

- The correction does not affect any original conclusions.
- The correction shows that SC's FID+ performance is better than previously reported: it improves from 12/24 cases to 17/24. That said, we reiterate that FID+ remains unreliable and should not be over-interpreted in SI-GNN evaluation.

Best,
Authors

---

### Author Response · Authors · 2025-12-03
**Summary Statement**

Below, we provide a concise summary for AC regarding the reviewer discussion.

**(1) How we addressed the reviewers' initial concerns**
- Reviewer 25Yx (score 8): Their minor presentation concern (color choices in Table 1) was fully addressed.
- Reviewer 7o1e (score 8): We added the requested encoder-freezing ablations and statistical significance tests; the reviewer confirmed that all concerns were fully addressed.
- Reviewer AnRn (score 4): We added the requested FID+ results for GAT and CAL and clarified the relationship to Tai et al.; the reviewer stated that their concerns were addressed and raised their score over the acceptance bar.
- Reviewer hKUr (score 4): Although the reviewer did not respond before the discussion freeze, we addressed all their concerns in detail (methodology, dataset scale, theoretical guarantee, evaluation metrics, and encoder-updating ablations). Several of these points overlapped with concerns raised by other reviewers, who indicated satisfaction with our explanations.

**(2) Reviewers' overall satisfaction after the discussion**
- Reviewer 25Yx was fully satisfied.
- Reviewer 7o1e wrote: *"You have addressed all my concerns, so I'll have to consider changing my rating to 10/10 if I see that the other reviewers’ concerns are also properly addressed."*
- Reviewer AnRn wrote: *"I am willing to raise my score over the acceptance bar given the changes you made."* We also responded to their follow-up comment before the discussion freeze, though no further exchange was possible.
- Reviewer hKUr did not respond before the freeze.

---

### Meta-Review · Area_Chair_Dom1 · 2026-01-11

**Summary:**

This paper addresses a key inconsistency issue in Self-Interpretable Graph Neural Networks (SI-GNNs). Typically, the training objective does not align with evaluation metrics such as faithfulness. To resolve this inconsistency, the authors propose a "Self-Consistency" (SC) training strategy. The core of this strategy is quite straightforward: a faithful explanation must remain consistent. Therefore, if we feed the explanation back into the model, we should obtain the same result. The method relies on a two-stage training process. The system first freezes the encoder and then fine-tunes the explainer using a specific SC loss.

Reviewers generally appreciated the approach's simplicity. However, initially, some reviewers raised several doubts regarding novelty and specific technical choices, including issues such as the rationality of the frozen encoder and the size of the dataset. The authors have responded to these questions through comprehensive rebuttals. They have introduced new ablation studies and statistical significance tests to prove their viewpoints.

**Reviewer Concerns:**

Reviewer AnRn mainly questioned the innovation of this paper, asking whether there is a fundamental difference between the proposed SC and the explanation integration (EE). In response, the author stated that there are indeed significant differences between the two. EE focuses on resolving inconsistencies between different models, while SC is used to address inconsistencies within a single model. Additionally, since SC is trained on a single model, it is much more efficient than EE. Reviewer AnRn accepted this explanation.

Reviewer AnRn also strongly recommended the inclusion of the FID+ metric. The authors supplemented these data. At the same time, they pointed out that FID+ has an OOD problem in SI-GNNs, and therefore is not as reliable as AUC. Additionally, the authors fixed a bug in their CAL implementation code, which improved the reported results.

Reviewers 7o1e and hKUr raised doubts about the decision to freeze the encoder, arguing that this design lacked validation. The authors responded with new ablation experiments. The results proved that freezing the encoder could lead to the most stable performance. Reviewer 7o1e was very satisfied with this proof.

Besides, reviewer 7o1e asked for statistical tests. The authors ran Mann-Whitney U tests. The results showed significant improvements in most cases.

Reviewer hKUr pointed out that the datasets are of relatively small size. The authors admitted this. However, this issue should be attributed to the entire field rather than the inadequacies in this particular paper. Reviewer hKUr also raised concerns about error propagation. The authors argued that this situation rarely occurs. They provided visual evidence to support this claim. Reviewer hKUr did not confirm if this answer was sufficient or not.

**Reviewer Scores:**

Reviewer 25Yx would likely keep the score at an 8. This reviewer was satisfied from the start. Small presentation fixes solidified their positive opinion.

Reviewer 7o1e would likely increase the score to a 9 or 10. After the release of new data on encoder freezing and the results of the significance test, they explicitly stated that they would raise the score to the full 10 points.

Reviewer AnRn would likely raise the score to a 6 or 7. After the rebuttal, they promised to move the score over the acceptance bar.

Reviewer hKUr did not respond to the rebuttal. However, the authors successfully conducted the requested ablation studies. These studies addressed the main critique regarding the encoder. So, I think the reviewer would likely raise the score to a 6.

---

### Decision · Program_Chairs · 2026-01-26

Accept (Poster)